# Enzymatic degradation of RNA causes widespread protein aggregation in cell and tissue lysates

Johan Aarum[1],[*],[†] ID, Claudia P Cabrera[2] ID, Tania A Jones[1] ID, Shiron Rajendran[1] ID, Rocco Adiutori[1], Gavin Giovannoni[1] ID, Michael R Barnes[2] ID, Andrea Malaspina[1] & Denise Sheer[1],[**] ID

## Abstract

**Most proteins in cell and tissue lysates are soluble. We show here that in lysate from human neurons, more than 1,300 proteins are maintained in a soluble and functional state by association with endogenous RNA, as degradation of RNA invariably leads to protein aggregation. The majority of these proteins lack conventional RNA-binding domains. Using synthetic oligonucleotides, we identify the importance of nucleic acid structure, with single-stranded pyrimidine-rich bulges or loops surrounded by double-stranded regions being particularly efficient in the maintenance of protein solubility. These experiments also identify an apparent one-to-one protein-nucleic acid stoichiometry. Furthermore, we show that protein aggregates isolated from brain tissue from Amyotrophic Lateral Sclerosis patients can be rendered soluble after refolding by both RNA and synthetic oligonucleotides. Together, these findings open new avenues for understanding the mechanism behind protein aggregation and shed light on how certain proteins remain soluble.**

**Keywords** motor neurone disease; neurodegeneration; phase transition; protein precipitation; ribonuclease

**Subject Categories** RNA Biology; Translation & Protein Quality

See also: **A Begeman *et al*** (October 2020)

## Introduction

Under physiological conditions, the majority of cellular proteins exist as soluble and folded entities. After protein synthesis, a complex machinery of cellular chaperones facilitates correct protein folding. Misfolded proteins are then identified and either corrected or, if the correction has failed, removed through autophagy or by the proteasome (Rubinsztein, 2006; Buchberger *et al*, 2010). Failure to resolve the misfolded state of proteins can lead to pathological accumulation and deposition of insoluble protein aggregates, as seen in many neurodegenerative diseases. However, aggregation *per se* is not necessarily a pathological phenomenon as various aspects of this process are also part of normal cellular physiology (Kaganovich *et al*, 2008; David *et al*, 2010; Wallace *et al*, 2015; Walther *et al*, 2015). For example, several proteins, in particular RNA-binding proteins, are recruited into semi-soluble, non-membrane encapsulated organelles such as stress granules and p-bodies through the process of liquid–liquid phase separation (Brangwynne, 2011). These, and similar structures, are not known to form pathological aggregates as the individual components are normally dynamically exchanged with the surrounding environment (Andersen *et al*, 2005; Kedersha *et al*, 2005; Spector & Lamond, 2011; Decker & Parker, 2012). However, there are several *in vitro* examples that liquid–liquid phase separation of proteins can form stable aggregates reminiscent of structures found in disease (Lin *et al*, 2015; Babinchak *et al*, 2019; Narayanan *et al*, 2019; preprint: Ray *et al*, 2019). Some of the key factors in liquid–liquid phase separations are just beginning to emerge and include, for example, the presence of low-complexity regions and/or unstructured regions in participating proteins (Kato *et al*, 2012; Aguzzi & Altmeyer, 2016; Bergeron-Sandoval *et al*, 2016; Wu & Fuxreiter, 2016). RNA also appears to play an important role in the formation of these structures. Indeed, RNA itself undergoes phase transition (Jain & Vale, 2017) and has been shown to promote phase transitions of several proteins (Lin *et al*, 2015; Molliex *et al*, 2015; Zhang *et al*, 2015), including Tau (Zhang *et al*, 2017), and also to inhibit protein aggregation, most notably of Fused in Sarcoma, FUS and TDP-43 (Shelkovnikova *et al*, 2014; Burke *et al*, 2015; Mann *et al*, 2019). Recently, Maharana *et al* (2018) showed that for several prion-like proteins, including FUS, these disparate effects can be explained by the ratio of protein to RNA, where excess of RNA promotes solubility and decreased amount of RNA induces phase transition. Similar results have also been observed for p53 and the prion protein (Kovachev *et al*, 2017, 2019).

Most neurodegenerative diseases are associated with aggregation of several proteins (Xia *et al*, 2008; Bai *et al*, 2013) and, curiously, the same proteins are frequently found aggregated across multiple diseases. For example, aggregated alpha-synuclein, a hallmark of

1  Barts and The London School of Medicine and Dentistry, Blizard Institute, Queen Mary University of London, London, UK
2  Barts and The London NIHR Cardiovascular Biomedical Research Centre, William Harvey Research Institute, Queen Mary University of London, London, UK
   *Corresponding author. Tel: +46 8 585 813 09; E-mail: johan.aarum@sll.se
   **Corresponding author. Tel: +44 20 7882 2595; E-mail: d.sheer@qmul.ac.uk
   †Present address: Department of Clinical Microbiology, Karolinska University Hospital, Stockholm, Sweden

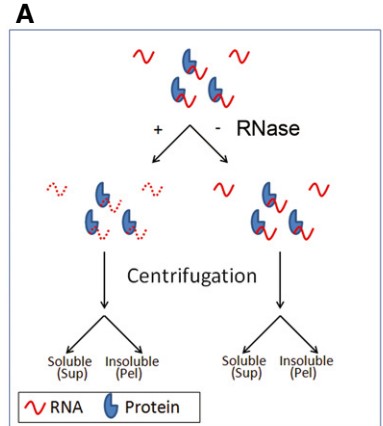
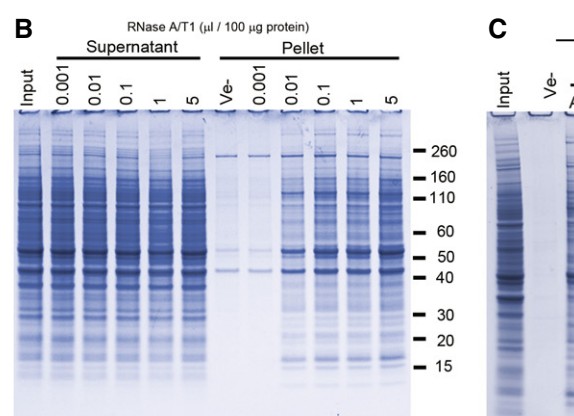

**Figure 1. Enzymatic degradation of RNA causes protein aggregation.**

A   Diagram showing the experimental design.

B   SDS–PAGE analysis of soluble (Supernatant) and insoluble proteins (Pellet) from human neurons after treatment for one hour with a mixture of RNase A and RNase T1 (A/T1), or vehicle (Ve-).

C   Protein aggregation (Pellet) after incubation with different ribonucleases or DNase I in the presence of either EDTA or $Mg^{2+}$. Ribonucleases used were RNase A (A), RNase T1 (T1), a mixture of RNase A and RNase T1 (A/T1), RNase 1f (1f) and RNase V1 (V1).

Parkinson's disease, can also be found in Alzheimer's disease (Hamilton, 2000; Mandal *et al*, 2006), and aggregated TDP-43, a hallmark of Amyotrophic Lateral Sclerosis (ALS), can also be found in both Alzheimer's disease and Parkinson's disease (Higashi *et al*, 2007; Nakashima-Yasuda *et al*, 2007). This indicates that a common factor might control the soluble state of these proteins. Altered RNA metabolism is a recurring theme in several neurodegenerative disorders (e.g. Ramaswami *et al*, 2013; Conlon & Manley, 2017; Liu *et al*, 2017). In ALS and Frontotemporal Dementia, this is supported by disease-causing genetic mutations in several RNA-binding proteins as well as by the generation of potentially toxic RNA species (Sreedharan *et al*, 2008; Kwiatkowski *et al*, 2009; Vance *et al*, 2009; Fratta *et al*, 2012; Kim *et al*, 2013; Haeusler *et al*, 2014). However, the link between altered RNA metabolism and protein aggregation *per se* is largely unexplored. Here, we show that RNA itself is critical in maintaining solubility of several disease-associated and other aggregation-prone proteins in cell and tissue lysates. These findings expand on previous studies by indicating a more general role for RNA, beyond its effect on RNA-binding proteins, in maintaining solubility of certain proteins.

## Results

### Enzymatic degradation of RNA causes protein aggregation

Within a few minutes after adding RNase to a clear cell lysate, it becomes opaque (Movie EV1), suggesting precipitation of material. To investigate if the precipitate contained proteins, we treated lysates prepared from human neurons and mouse brain cortex with a mixture of RNase A and RNase T1 (A/T1) and identified any aggregated proteins by gel electrophoresis (the experimental outline is shown in Fig 1A). RNase treatment caused a concentration-dependent aggregation of proteins from both samples (Figs 1A and EV1A) and was accompanied by a decrease in both the amount and size

distribution of RNA in the lysate (Fig EV1B). RNase A/T1 treatment for one hour at 37°C consistently precipitated $10\% \pm 1\%$ ($n = 4$) of the amount of input proteins from both human neurons and mouse brain, a fraction that did not increase by prolonged RNase treatment.

The specificity of the ribonuclease was not important for the overall efficiency, as the single-stranded ribonucleases RNase A, T1 and 1f showed similar efficiency to the RNase A/T1 mixture (Fig 1C). RNase V1, which is specific for double-stranded RNA, also caused protein aggregation but only if EDTA was omitted from the buffer and replaced with $Mg^{2+}$, (Fig 1C). RNase V requires $Mg^{2+}$ for its activity. However, DNase I failed to cause protein aggregation in the presence of either EDTA or $Mg^{2+}$ (Fig 1C). No proteins above background (i.e. in samples without added nucleases) were aggregated when increasing amounts of a ribonuclease inhibitor were added to the lysate together with RNase A (Fig EV1C), or when enzymatically or chemically (NaOH) degraded RNA was added to the lysate (Fig EV1D).

### Mass spectrometry identification of aggregated proteins

We used liquid chromatography tandem mass spectrometry (LC-MS/MS) to identify the proteins which are aggregated by degradation of RNA in human neuronal cell lysates. More than 1,300 aggregated proteins were found to be common to two biological replicates (Dataset EV1). Gene ontology analysis of the data set against a total proteome (approximately 6,600 proteins) from human neurons (Song *et al*, 2019) using PANTHER indicates an over-representation of protein-containing complexes (FDR $q = 1.22 \times 10^{-18}$), proteins involved in translation (FDR $q = 7.59 \times 10^{-12}$), RNA binding (FDR $q = 1.97 \times 10^{-13}$) and heterocyclic compound binding (FDR q = $5.93 \times 10^{-18}$; Fig EV2A–C and Dataset EV1). Since mass spectrometry analysis of complex protein mixtures is more likely to identify more abundant proteins than proteins present in low quantities, we reanalysed our data set using the proteins with higher than average relative abundance in Song *et al* (2019) as background. Similar enrichment of GO terms was

observed (Dataset EV1), suggesting that the proteins that aggregate upon RNase treatment are not random. Recently, unstructured, low-complexity regions in several RNA-binding proteins have been shown to mediate protein phase transition (Kato *et al*, 2012). However, both low-complexity (LC) and unstructured (US) regions are significantly under-represented in our data set (Fig EV2D and E).

The aggregated proteins include several proteins that are associated with neurodegenerative disease or other proteinopathies, such as huntingtin (HTT), TDP-43, Gelsolin, Lysozyme, the heterogeneous nuclear ribonucleoproteins A2B1 (HNRNPA2B1), HNRNPA1, and in one of the LC-MS/MS replicates, the valosin-containing protein, VCP (Haltia *et al*, 1990; Pepys *et al*, 1993; DiFiglia *et al*, 1997; Hirabayashi *et al*, 2001; Watts *et al*, 2004; Neumann *et al*, 2006; Kimonis *et al*, 2008; Kim *et al*, 2013). We used Western blot to validate the mass spectrometry data and to investigate the solubility of other aggregation-prone proteins associated with proteinopathies. HTT, neurofilament heavy chain (NF-H), Tau (MAPT), FUS, TDP-43, HNRNPA1, HNRNPD, RPL7 and actin (ACTB, found aggregated in Hirano bodies in several neurodegenerative diseases; Hirano, 1994), were selectively aggregated upon RNase A/T1 treatment of human neuronal lysates (Fig 2A). However, the solubility of poly A binding protein, PABP, an abundant RNA-binding protein not identified in aggregates by mass spectrometry, was unaffected by RNase treatment (Fig 2A). Similar results were obtained using tissue lysate prepared from mouse cortex (Fig EV3). We also detected an approximately 40 kDa Amyloid beta (Aβ)−immunoreactive band in the pellet of RNase-treated lysate (Fig 2A), possibly representing Aβ oligomers (Walsh *et al*, 1997; McLean *et al*, 1999) formed from Aβ generated in intracellular vesicles (Rajendran *et al*, 2006). Since the molecular weight of this band is larger than expected, we also examined the aggregation of Aβ in cell lysates prepared from HEK293T cells expressing Aβ fused to GFP. This fusion protein aggregated upon degradation of RNA, while no aggregation was observed for GFP itself (Fig 2B).

Inhibiting the added RNase activity with an RNase inhibitor abolished the aggregation of HTT, NF-H and TDP-43 (Fig 2C).

## RNA is required for maintaining the non-aggregated state of renatured proteins

We next denatured proteins aggregated by enzymatic degradation of RNA in 6M guanidine hydrochloride (GuHCl) and then attempted to renature them in the presence or absence of total RNA without any prior size fractionation (experimental outline in Fig 3A). We hereafter refer to this process, i.e. GuHCl denaturation of RNase-aggregated proteins followed by refolding in the presence of various additives, as renaturation. After removal of GuHCl, proteins remained soluble in the presence of RNA in an RNA concentration-dependent manner, while the majority of proteins without RNA re-aggregated (Fig 3B and C). Remarkably, enzymatic degradation of RNA from the soluble fraction (Sup 1) after renaturing in the presence of RNA caused the proteins to re-aggregate (Fig 3C, Pel 2). The same principles were also observed for individual proteins (Fig 3D).

Other nucleic acids, including total *E. coli* RNA and human genomic DNA, efficiently prevented protein aggregation while

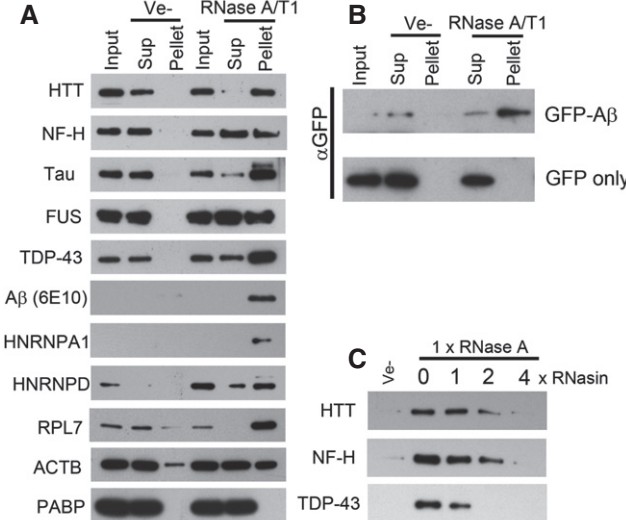

**Figure 2. Enzymatic degradation of RNA induces aggregation of proteins associated with neurodegenerative diseases.**

A, B   Western blot analysis of soluble and aggregated proteins after RNase treatment of lysate from human neurons (A), or HEK293T cells expressing GFP-Aβ or GFP (B).
C       Effect of an RNase A inhibitor (RNasin) on RNase A mediated protein aggregation in neuronal cell lysate.

neither yeast tRNA nor heparin at the same amounts, did (Fig EV4A). The effect of DNA is surprising since our data show that proteins in cell lysate are maintained in a soluble state by intact RNA and not by DNA (Fig 1C).

It is possible that the degradation of RNA causes the aggregation of a few proteins that then co-sequester and aggregate with many other proteins. To investigate this hypothesis, we therefore purified recombinant TDP-43, which forms inclusion bodies in *E. coli* (Furukawa *et al*, 2011; Capitini *et al*, 2014), under denaturing conditions (6 M guanidine hydrochloride) and assessed the proportions of soluble and insoluble TDP-43 when renatured in the presence or absence of RNA. If co-sequestering did occur, then the presence or absence of RNA should not affect the solubility of TDP-43. However, we only obtained soluble TDP-43 when renatured with RNA (Fig EV4B). Therefore, at least for TDP-43, the aggregation depends solely on the availability of RNA, confirming previous studies that TDP-43 aggregation is enhanced by lack of RNA interactions (Elden *et al*, 2010; Pesiridis *et al*, 2011).

## RNA is required for protein activity

Correct folding is required for the proper function of proteins. To investigate whether the renatured proteins were functional and not just solubilised, we assessed the ATP-hydrolysing activity of ATP-binding proteins, which represent a large proportion of the aggregated proteins (18%, 240/1312). As the ability to bind ATP depends on the presence of conserved structural motifs (Walker *et al*, 1982), we first investigated if the ability to bind ATP was restored in the presence or absence of RNA. Proteins from human neurons bound to ATP only in the presence of RNA (Fig EV4C), and LC-MS/MS

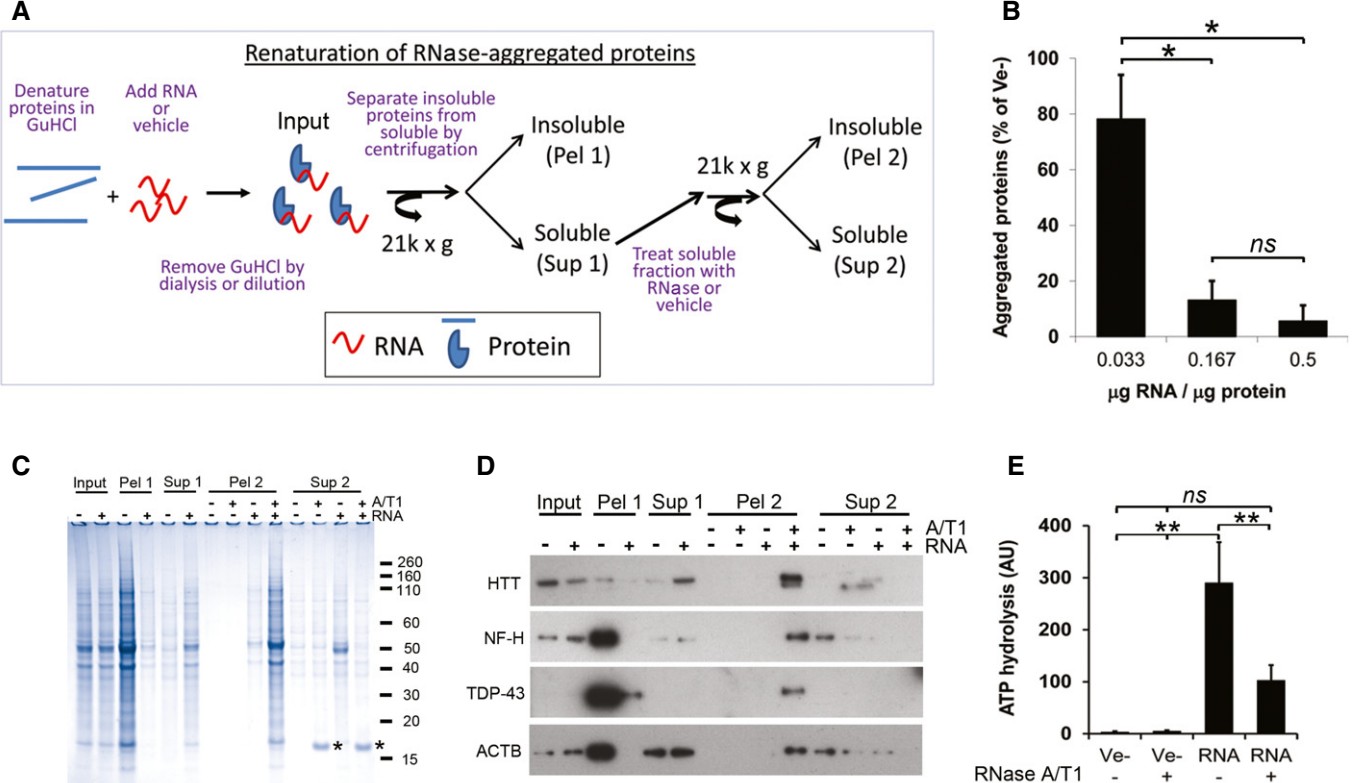

**Figure 3. RNA is required for renaturation and function of proteins aggregated by enzymatic degradation of RNA in human neuronal lysates.**

A Diagram showing the renaturing assay.
B Effect of RNA/protein ratio on protein aggregation after renaturing.
C Coomassie-stained gel of soluble (Sup 1/2) and aggregated (Pel 1/2) proteins after removal of GuHCl in the presence (+) or absence (−) of total RNA. Asterisk (*) denotes added RNase A.
D As in (C) but analysed by Western blot.
E ATP-hydrolysing activity of proteins renatured with total RNA or buffer (Ve−) after removal of GuHCl.

Data information: All experiments were performed with total RNA. Data in (B and E) are expressed in arbitrary units (AU) and represent the mean ± s.d of two (B) or three (E) independent experiments. **$P < 0.01$, *$P < 0.05$ by *post hoc* ANOVA.

characterisation of these showed a clear enrichment of ATP-binding proteins (58/143, Dataset EV2). RNA-renatured proteins hydrolysed 100-times more ATP than proteins without RNA (Fig 3E), and concomitant degradation of RNA by the addition of RNase A/T1 hampered this activity (Fig 3E). Similar results were obtained with aggregated proteins from Jurkat cells (Fig EV4D).

**Identification of RNA sequences that are associated with soluble proteins**

To identify transcripts associated with soluble proteins in human neuronal lysates, we performed native RNA immunoprecipitation (RNA-IP) for Aβ and NF-H, followed by sequencing. The majority of associated transcripts were derived from protein-coding genes. 471 different transcripts associated with Aβ and 123 with NF-H, respectively (Dataset EV3). This RNA-IP experiment indicates which transcripts are associated with the proteins in cell lysate but give little information as to which regions, e.g. UTRs or coding regions, are involved in the interactions. To increase the mapping resolution, we therefore performed RNA immunoprecipitation for Aβ and NF-H, as

well as Tau, on proteins renatured with pre-fragmented total Jurkat RNA (~100 nucleotides).

For all three proteins, binding peaks were mainly observed in exons (33–53% of all binding sites), in introns (15–21%) and in 3′-UTRs (6–10%), respectively (Fig 4A, GEO GSE99127). As in the native RNA-IP experiments on Aβ and NF-H, the vast majority of the binding sites derived from exons and introns came from protein-coding transcripts (Fig 4A). Between a fifth (NF-H 25/123) and a third (Aβ 180/471) of all coding transcripts in the native RNA-IP experiments for A and NF-H were also represented by binding peaks in the renatured samples (Dataset EV3). These experiments indicate that soluble Aβ, NF-H and Tau are associated with a variety of transcripts, particularly with the coding regions of these. To further deduce the principles governing this phenomenon, we next turned to DNA, which, because of the similar renaturation capacity to RNA (Fig EV4A), represents a convenient and easily manipulated model system.

First, to identify DNA sequences capable of renaturing aggregated proteins, we performed two sets of isolation and sequencing experiments using proteins renatured with pre-fragmented genomic

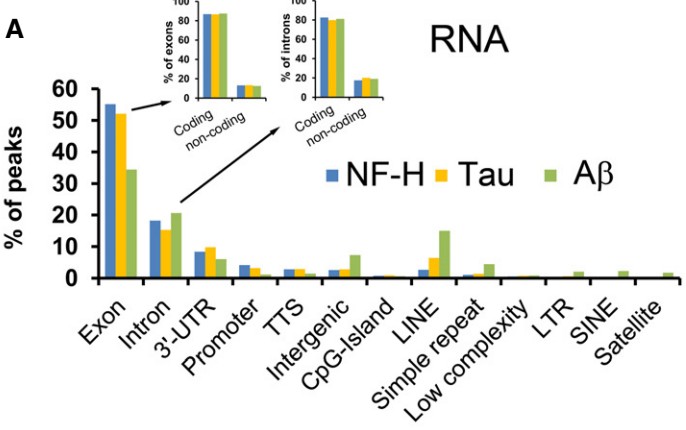

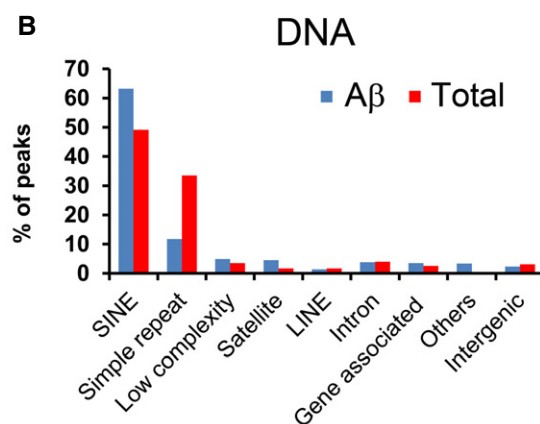

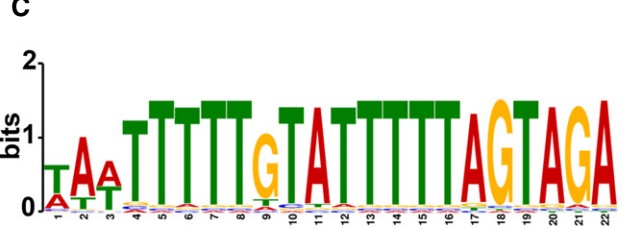

**Figure 4. Features of solubilising nucleic acids.**

A  Genomic attributes of RNA associated with renatured NF-H, Tau or Aβ. Inset shows which fractions of the peaks that derive from exons or introns have their origin in coding or non-coding transcripts, respectively.

B  Characterisation of DNA associated with soluble, renatured Aβ or all proteins (Total), captured either by immunoprecipitation (Aβ) or absorption to membranes (Total).

C  Sequence logo of the computationally identified motif (M1) in the solubilising genomic DNA of the Total samples.

DNA (approximately 300 bp). In the first set, we aimed to isolate all soluble proteins (~1,300) and their associated DNA fragments by capturing them on nitrocellulose membranes. These samples are hereafter referred to as "Total". In the second set of experiments, we isolated Aβ and its associated DNA fragments by immunoprecipitation. For both samples, the majority (Aβ ~86%, Total ~90%) of protein-associated fragments were from repetitive DNA regions, in particular Short INterspersed Elements (SINEs) (including *Alu* repeats, 63 and 49%, respectively) and from regions containing simple repeats (11 and 33%, Fig 4B, GEO GSE99127).

We next tried to identify common sequence motifs in these data sets. We found a pyrimidine-rich motif (Motif 1, M1, Figs 4C and EV5A) in the "Total" DNA data set that was similar to motifs found in both the RNA and the DNA samples associated with renatured and immunoprecipitated Aβ (Fig EV5B).

### Complementary strands are required for efficient protein renaturation

We tested the M1 motif in a 4-repeat form, i.e. M1 × 4, and a control oligonucleotide, Motif 2, M2, randomly generated to have a similar proportion of G/C (Fig EV5A), for their capacity to renature the aggregated proteins.

As single-stranded (ss) DNA sequences, none of the oligonucleotides supported overall renaturation (Fig 5A). However, when both the forward and the reverse (For/Rev) oligonucleotides were used together they efficiently supported renaturation in a concentration-dependent manner (Figs 5A and EV5C), with a theoretical stoichiometry of ~1:1 (based on the assumption that the average molecular weight of a protein is 50 kDa). The M1 × 4 oligonucleotide was twice as potent in renaturing the proteins than the M2 control, indicating a sequence preference but not a strict sequence requirement. Similar results were observed for Actin, HTT, TDP-43 and RPL7 (Fig 5B). Nucleolin (NCL), however, was equally renatured by all configurations (Fig 5B). Thus, although the majority of the tested proteins in our mixture have a preference for complementary oligonucleotides, individual differences do exist.

### Both single- and double-stranded regions are needed for efficient protein renaturation

We next investigated how the size of the DNA oligonucleotides affects their renaturation capacity by decreasing the number of motif repeats. We found that only oligonucleotides with at least three motif repeats (e.g. Mx3) could efficiently renature the proteins (Fig 5C), indicating that there could be a size requirement. However, experiments on single-stranded segments formed after our standard nucleic acid heat denaturation and rapid cooling step (see Materials and Methods) suggested that there might also be a structural requirement.

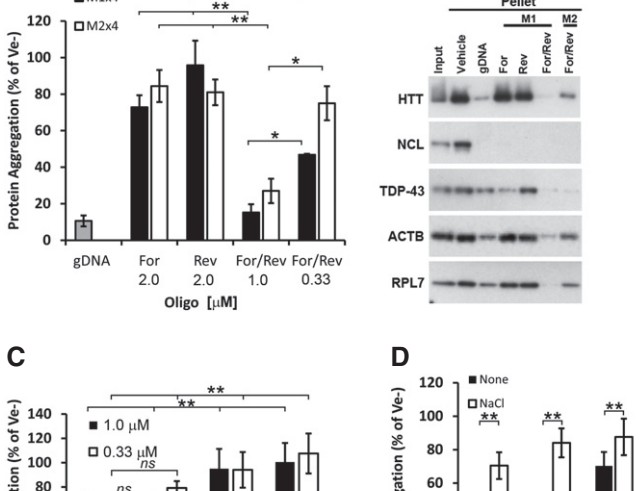

**Figure 5.  Renaturing characteristics of selected motifs.**

A   Proportion of aggregated proteins after renaturation with either single-stranded (Forward or Reverse) or complementary strands (For/Rev) of the selected M1x4 or the control oligonucleotide, M2x4.

B   Western blot analysis of proteins renatured with either single-stranded or double-stranded M1x4 or M2x4 DNA oligonucleotides.

C   The effect on protein re-aggregation from varying the number of DNA motif repeats when complementary oligonucleotides are given together.

D   Protein aggregation after renaturation with the M1x4 DNA oligonucleotides pre-annealed with 100 mM NaCl or vehicle (None).

Data information: Data are expressed as a fraction of vehicle (Ve−) and represent the mean ± s.d of three to four independent experiments.

**P < 0.01, *P < 0.05 by post hoc ANOVA.*

To test this possibility, we performed a pre-annealing step in the presence of 100 mM NaCl, which promotes the normal double-stranded form. Only oligonucleotides prepared without NaCl (non-annealed) were able to renature the proteins (Fig 5D). The amount of NaCl used in the pre-annealing reaction had no effect on protein solubility.

## Efficient nucleic acid-mediated protein renaturation requires looped, pyrimidine-rich single-stranded regions

Since the M1 oligonucleotide is pyrimidine-rich, we next tested whether there is a preference for pyrimidines in the ss region by examining the renaturing efficiency of oligonucleotides where the ss regions were either composed of 28 consecutive pyrimidines or purines (Fig 6A). When assessed for their renaturing capacity, oligonucleotides with a ss region of pyrimidines were superior to the same oligonucleotides containing a ss region of purines (Fig 6B).

We then examined a series of oligonucleotides, all containing similar double (ds) and single-stranded regions but positioned at different locations along the DNA (see second panel of Fig 6C for their theoretical structures). Of these, the most efficient configuration was the ds/ss/ds forms, in particular the 3x-loop or 3x-bulge oligonucleotides (Fig 6C).

## Synthetic DNA oligonucleotides can replicate the effect of endogenous RNA in cell lysates

We next examined whether these oligonucleotides could prevent protein aggregation caused by enzymatic RNA degradation in cell lysates. Both single-stranded and, in particular, complementary DNA oligonucleotides almost completely prevented protein aggregation following RNase A/T1 treatment (Figs 6D and EV5D).

The unexpected finding that the single-stranded oligonucleotides were also capable of preventing protein aggregation, albeit less than complementary DNA oligonucleotides (Fig 6D), indicates that they may complement with endogenous RNA in the lysate, forming structural RNA/DNA hybrids. However, RNase H treatment of the lysate had little or no effect (Fig EV5D), suggesting that any hybrids present are protected.

## Renaturation of protein aggregates from Amyotrophic Lateral Sclerosis patient brain

Finally, we isolated aggregated proteins from brain tissue samples of two patients with ALS and investigated whether NF-H within these proteins could be renatured (after denaturation with GuHCl) using nucleic acids. We examined neurofilament (NF) as it is found in protein aggregates of several neurodegenerative diseases (reviewed in Didonna & Opal, 2019), including ALS (Mendonça *et al*, 2005) and was readily detected by Western blot in the

**Figure 6.  Structural characteristics of protein-solubilising oligonucleotides and renaturation of ALS brain-derived protein aggregates.**

A   Cartoons of ds/ss/ds oligonucleotides having either pyrimidines (T or C) or purines (A) in the ss region.

B   Proportion of protein aggregation following renaturing with the ds/ss/ds oligonucleotides shown in (A).

C   Renaturing capacity of structurally different oligonucleotides. The diagrams on the right show a theoretical structure of each oligonucleotide. All oligonucleotides, except the 3x-loop and 3x-bulges, contain a stretch of 30 Ts in the single-stranded regions and the same sequences (15 nucleotides each) in the double-stranded regions. The 3x-loops and 3x-bulges oligonucleotides have 3 stretches of 9 Ts and the same sequence in the ds-regions.

D   Proportion of protein aggregation in Jurkat cell lysate supplemented with various amounts and configurations of the M1x4 or M2x4 DNA oligonucleotides.

E   Insoluble proteins from two ALS brain tissues were chemically denatured in guanidine hydrochloride and treated with either buffer (Vehicle), total RNA from Jurkat cells or the complementary strands of the M1x4 DNA oligonucleotides (M1 F/R) (Pellet 1, top panel). After removal of GuHCl, the soluble fraction from these samples was treated with RNase and any aggregated proteins (Pellet 2) analysed by Western blot (middle panel). Remaining supernatants were then treated with Benzonase to degrade any remaining nucleic acids and aggregated proteins collected by centrifugation and analysed by Western blot (bottom panel). Proteins aggregated by enzymatic RNA degradation in Jurkat cell lysates, which do not contain NF-H, were used as a control.

Data information: Data in (B and C) are expressed as a fraction of vehicle (Ve−), while data in (D) are expressed as the amount of aggregation observed without any oligonucleotides present (A/T1). All bars represent the mean ± s.d of two to four independent experiments. **P < 0.01 by post hoc ANOVA.*

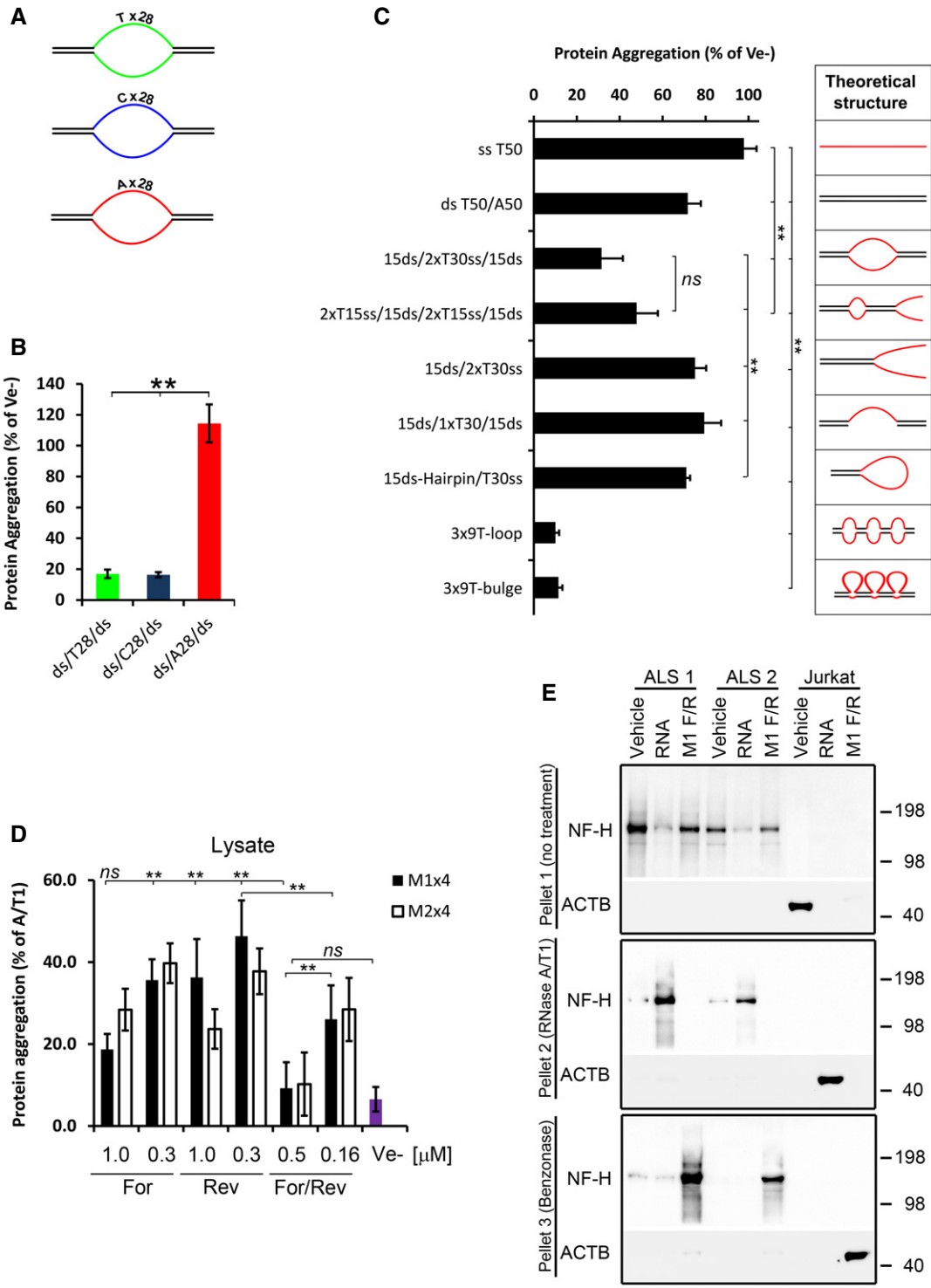

**Figure 6.**

starting brain lysates while we could not detect TDP-43 in the same, possibly due to low amount. Total RNA and the M1x4 oligonucleotides efficiently renatured NF-H from the ALS samples, while samples treated with vehicle show pronounced re-aggregation (Fig 6E, top and second panel). No NF-H signal was observed in the Jurkat cell samples (used as control) as these cells do not

express NFs. These were instead positive for Actin (Fig 6E). Ribonuclease treatment of the soluble fractions (Sup 1) of proteins renatured with RNA caused re-aggregation of NF-H and Actin, but had no effect on the proteins renatured with the M1x4 DNA oligonucleotides, which remained soluble even after 16 hour at 37°C (Fig. 6E, middle panel). However, NF-H and actin in these

DNA-containing fractions were efficiently re-aggregated after a further treatment step with the non-selective nuclease, Benzonase (Fig 6E, lower panel).

## Discussion

Many proteins are known to bind RNA, for example the well-characterised hnRNP family of proteins involved in RNA splicing (Swanson & Dreyfuss, 1988). In our study, we build on previous findings that solubility of several RNA-binding proteins can be enhanced by RNA (Shelkovnikova *et al*, 2014; Sun *et al*, 2014; Burke *et al*, 2015; Kovachev *et al*, 2017, 2019; Maharana *et al*, 2018; Mann *et al*, 2019). We provide novel insights into how a wide range of proteins, many of which are not known to bind to RNA, are maintained in a soluble state *in vitro*. We show that this heterogeneous set of proteins is prevented from aggregation by association with RNA and, crucially, that the presence of intact RNA is required to maintain them in a soluble state in cell and tissue lysates (Figs 1–3). We also provide evidence that RNA is required for the functionality of some proteins, as exemplified by the pronounced difference in ATP-hydrolysing activity between proteins renatured in the presence or absence of RNA (Figs 3E and EV4C). Finally, we provide a tantalising link between these observations and human disease by demonstrating the efficient renaturation of insoluble proteins isolated from human ALS brain tissue.

Enzymatic RNA degradation causes the aggregation of a consistent set of proteins from neuronal cell lysates. Although there is a significant over-representation of RNA-binding proteins amongst them (394 of 1,312 proteins), the vast majority lack conventional RNA-binding domains. This raises at least two possible explanations for the variety of aggregated proteins. First, either most of the proteins associate with RNA through non-conventional RNA-binding regions or, alternatively, the degradation of RNA initiates the aggregation of a few RNA-associated proteins (i.e. seed proteins) which then sequester a large number of other proteins, irrespective of their association with RNA. While a combination of these two scenarios is likely, and both are supported by the literature (Olzscha *et al*, 2011; Castello *et al*, 2012), our data favour the first possibility as the major driving force of protein aggregation mediated by enzymatic RNA degradation. This is because *in vitro* renaturing of the proteins in the presence of the correct nucleic acids efficiently maintains solubility of the vast majority of them after removal of guanidine hydrochloride (Figs 3, 5 and 6). Should, however, a small set of RNA-binding seed proteins cause the aggregation of the majority, only these would be expected to require RNA for renaturation, while most of the other proteins should be unaffected (if the RNA-binding seed proteins are not also efficient chaperones).

Fundamentally, and based on the generalisation that the average molecular weight of a protein is 50 kDa, we find that in order to achieve efficient refolding, the nucleic acids and the proteins need to be present in equimolar quantities [i.e. 30 μg of proteins (~600 pmol) is efficiently renatured by 600 pmol of oligonucleotides, see e.g. Fig 5]. Such a stoichiometry suggests that each protein needs to be associated with one oligonucleotide to maintain solubility. These figures are in line with what has been observed for other aggregate-prone proteins such as the prion protein and FUS (Maharana *et al*, 2018; Kovachev *et al*, 2019). If seed-driven co-

precipitation was a major force, a much lower oligo-to-protein ratio would be expected, as only the few seed proteins need to be associated with the nucleic acid.

The cellular proteome contains many polyanion-binding proteins which are critical for proper function of cells and organisms (reviewed in ref. Jones *et al*, 2004). With respect to protein stability and aggregation, the effect of polyanions varies, sometimes in opposing ways, for different proteins, or even between different polyanions on the same protein. As an example of these varying and conflicting effects, aggregation and pathological conversion of the prion protein is stimulated by the presence of RNA and possibly by endogenous proteoglycans, but is inhibited by heparin and exogenous forms of proteoglycans (Caughey & Raymond, 1993; Wong *et al*, 2001; Deleault *et al*, 2003; Vieira *et al*, 2014). Similarly, FUS aggregation is either inhibited or promoted by the presence of presumably different species of RNA (Shelkovnikova *et al*, 2014; Lin *et al*, 2015). Tau aggregation on the other hand has been reported to be stimulated by RNA (Kampers *et al*, 1996; Zhang *et al*, 2017), while the aggregation of Aβ has been reported to be inhibited (Mathura *et al*, 2005; Takahashi *et al*, 2009), a finding also confirmed here. Thus, it is clear that polyanions, at least *in vitro*, have diverse and sometimes paradoxical effects on protein structure and solubility, possibly depending on the protein/polyanion concentration (Kovachev *et al*, 2017, 2019; Maharana *et al*, 2018).

Consistent with these findings and those in the accompanying paper (Begeman *et al*, 2020), our data show a varying capability of different nucleic acids to promote protein renaturation and maintain solubility and, strikingly, this effect is not directly determined by sequence, but primarily by structure (Figs 5 and 6). This agrees with recent findings that highly structured RNAs have more interactions with proteins than RNAs with low structural complexity (Sanchez de Groot *et al*, 2019). In our study, the most efficient renaturing and aggregate-preventing nucleic acids consist of pyrimidine-rich loops or bulges interspersed by stretches of double-stranded regions, e.g. nucleic acids with a ds/ss/ds configuration (Figs 6B and C). In addition, the M1 oligonucleotide prevents protein aggregation when both strands are provided, while the same amount (in moles) of ss oligonucleotides (i.e. twice the amount of the ds oligo) fails to do so (Figs 5A and B). Similarly, when the two strands are pre-annealed, i.e. seemingly forming a perfect ds strand, the renaturing effect is all but lost (Fig 5D). These results also suggest that the anionic charge provided by the nucleic acids (mainly by the phosphate backbone) cannot be its sole contribution, as the same amount of charge is provided by the two different configurations. This notion is further supported by the much higher efficiency observed for oligonucleotides with a high proportion of pyrimidines in the single-stranded region as compared to their purine-rich counterparts (Fig 6). Indeed, the preference for pyrimidines over purines suggests that the bases themselves, and not solely the negative sugar backbone *per se*, are important for the solubilising effect. Such structural dominance may also explain some of the conflicting effects observed with polyanions on protein solubility(Shelkovnikova *et al*, 2014; Lin *et al*, 2015). More generally, and as also suggested by others (Zhang *et al*, 2015), such disparate effects point to the intriguing possibility that some nucleic acids may not only prevent protein aggregation but may actually promote/induce the same.

The structure- and base-dependent principles summarised above explain the majority of the observed effects, but we do see

deviations from these rules for individual proteins. For example, while nucleolin clearly requires nucleic acids for efficient renaturation, similarly efficient renaturation is achieved with either single-stranded or looped oligonucleotides (Fig 5B). Thus, it is likely that other structures, besides the highly efficient looped/bulge configuration described here, can efficiently renature individual or groups of proteins. Indeed, sequences forming G-Quadruplex structures efficiently prevent and enhance protein folding, including in living *E. coli* cells (Begeman *et al*, 2020). Similarly, bacterial 23S ribosomal RNA and nucleic acid homopolymers (e.g. single-stranded poly Ts) have recently been shown to assist renaturation/refolding of a number of proteins, an effect suggested to derive from the nucleic acids functioning as non-protein molecular chaperones (Chattopadhyay *et al*, 1996; Sulijoadikusumo *et al*, 2001; Docter *et al*, 2016). Although our data support the general conclusions of these studies, we suggest that the solubilising nucleic acids are integral parts of these complexes with roles beyond traditional chaperones, whose effects are usually energy-dependent and mediated through transient interactions of importance mainly for initial folding. This view is backed by our finding that ATP-hydrolysing proteins require the continuous presence of RNA for activity *in vitro* and, similarly, enzymatic degradation of RNA from proteins renatured in the presence of RNA causes them to re-aggregate. Of note, all our renaturing experiments were performed in the absence of traditional energy sources such as ATP.

The main feature in solubilising genomic DNA is interspersed repeat elements, in particular SINE elements and low-complexity regions. This is true both for individual aggregation-prone proteins (Aβ) as well as for the majority of the proteins, i.e. Total (Fig 4B). While some of these may represent true associations, it is plausible that we are observing a pure *in vitro* effect. The argument for this is that similar, but divergent, repeats are likely to readily form the ds-ss-ds structures found here to be highly efficient in protein renaturation.

What is the nature of the protein–RNA associations? Following on from our arguments above, our data support a form of interaction where the permissive nucleic acid (RNA) is an integral but non-covalent part of certain proteins or protein complexes. To be biologically relevant, these arrangements are likely to be dynamic, with proteins and RNA being changeable, i.e. the same RNA may associate with different proteins and *vice versa*. Indeed, such interchangeability is supported by our *in vitro* sequencing and oligonucleotide characterisation studies as individual proteins are found associated with different nucleic acids, and different proteins are found associated with the same nucleic acids (Fig 4). Similar promiscuity is likely to exist also in cells (Maharana *et al*, 2018), as it is unlikely that the solubility of a particular protein is maintained only by any one particular RNA.

The protein–RNA associations are influenced by nucleic acid structure, as discussed above, but are also likely to be determined by post-translational modifications of the proteins. Indeed, the capacity of several RNA-binding proteins to undergo phase transition *in vitro* is inhibited by arginine methylation, e.g. FUS (Hofweber *et al*, 2018; Qamar *et al*, 2018) or serine phosphorylation, e.g. TDP-43 (Wang *et al*, 2018). More specifically, certain arginine/serine rich proteins (e.g. SRSF Protein Kinase 1) require either association with RNA or serine phosphorylation to be soluble in cell lysate (Nikolakaki *et al*,

2008). However, in addition to such post-translational modification, it is likely that associations with other proteins or macromolecules could replace the requirement for RNA. In support of this observation, for some of the proteins we examine here, like HTT and RPL7, RNA degradation causes substantial aggregation with little or no soluble protein left in the supernatant (Figs 2A and C). For others, e.g. NF-H, FUS and TDP-43, only a proportion appears to aggregate with varying amounts still detected in the soluble fraction, even after prolonged RNA degradation. This could also explain why we do not see any visible enrichment of aggregated proteins compared with the input when analysed on PAGE gels (Figs 1B and EV1A). However, the amount of proteins, both in terms of mass and numbers, combined with the poor resolution of PAGE gels makes direct comparisons of different bands problematic.

It is clear that RNA has several important biological roles besides its conventional function as a template for translation and assisting in protein synthesis. These include acting as post-transcriptional regulators (e.g. miRNA), as structural scaffolds (e.g. ncRNA and rRNA), as cofactors (primers for DNA replication) and as enzymatic entities, i.e. ribozymes. The data presented here suggest that RNA has an additional role in maintaining proteins in cell and tissue lysate in a soluble and presumably functional form.

Finally, our data indicate that alterations to RNA could contribute to pathological protein aggregation, providing a mechanistic rationale for the observed aggregation of the same proteins across multiple diseases.

# Materials and Methods

### Cells and ALS samples

Neurons were differentiated from human neural stem cells (hNP1 cells) by withdrawal of basic FGF for 6 days, as described (Jones *et al*, 2011). The majority (> 95%) of the cells differentiate into Map2- and β III-tubulin-positive cells within 6 days (Jones *et al*, 2011).

Jurkat T cells and HEK 293T (CRL-3216, ATCC) were maintained in RPMI 1640 (21875-034, Life Technologies) supplemented with 10% FCS (Life Technologies) and 1 x Penicillin-Streptomycin (15070063, Life Technologies). All cells were maintained at 37°C in 5% $CO_2$.

Cortices from day 16-21 C56BL mice were dissected at room temperature (RT), rolled on filter paper to remove most of the meninges and immediately frozen on dry ice and stored at −80°C until use.

The ALS brain samples were obtained from The Netherlands Brain Bank (NBB), Netherlands Institute for Neuroscience, Amsterdam (open access: www.brainbank.nl), under ethical permission 2009/148. All material was collected from donors from whom a written informed consent for a brain autopsy and the use of the material and clinical information for research purposes had been obtained by the NBB. Samples were obtained from the precentral gyrus of two ALS donors, both male, 62 and 71 years of age, with an ALS diagnosis confirmed neuropathologically by the detection of TDP-43 inclusions, particularly in the spinal cord. The study of proteins from human ALS tissue has been granted to AM by the London—City & East Research Ethics Committee, reference number: 09/H0703/27.

## Enzymes and reagents

RNase T1 (AM2280), RNase V1 (AM2275), RNase A/T1 cocktail (EN0551), DNase I (2222) and Yeast tRNA (15401-011) were from Thermo Fisher Scientific. RNase A (R4642), Sodium acetate (S7899), and EtOH were from Sigma. RNase 1f (M0243) was from New England Biolabs (NEB). Heparin (07980) was from Stemcell Technologies. Guanidine hydrochloride (BP178-1) was from Fisher Scientific.

## Preparation of cell-free lysates from neurons and mouse cortex

Differentiated neural stem cells were detached by trypsin (0.5%, Life Technologies) and collected in RPMI 1640 medium with 10% FCS (Life Technologies). Cells were pelleted by centrifugation and washed twice in ice-cold PBS (14190-094) before being lysed in four cell-pellet volumes of either Lysis Buffer 1 [20 mM Tris–HCl pH 7.5, 150 mM NaCl, 3 mM EDTA, 1% Triton X-100, 0.5% Na-Deoxycholate, 1X protease inhibitors cocktail (Roche), 1 mM DTT] or Lysis Buffer 2 [20 mM Tris–HCl pH 7.5, 150 mM NaCl, 1.5 mM MgCl$_2$, 1% Triton X-100, 0.5% Na-Deoxycholate, 1X EDTA-free protease inhibitors cocktail (Roche), 1 mM DTT]. Most experiments were performed in Lysis Buffer 1, except when DNase I or RNase V1 treatment was performed (Fig 1D), in which case Lysis Buffer 2 was used. Lysed cells were sonicated (Bioruptor, Diagenode) at maximum setting for 5 s on ice and centrifuged at 21,000 g for 30 min at 4°C. The supernatant was filtered through a 0.1 μm syringe filter (Santa Cruz, sc-358809) into new tubes and the protein concentration determined with the BCA kit (Thermo Fisher) according to the manufacturer's instructions. Lysates were diluted in Lysis Buffer-1 or Lysis buffer-2 to 2–4 μg/μl, and treated as described below.

Mouse cortical tissue was thawed on ice and disrupted in cold PBS using a 1-ml pipette tip. Disrupted tissue was washed 3 times in PBS before being lysed in Lysis Buffer 1 and prepared as described above for human neurons.

## Ribonuclease treatment and isolation of aggregated proteins

Typically, 150–400 μg cell lysates at 2–4 μg/μl were mixed with indicated amounts of ribonucleases, DNase I or Vehicle (50% Glycerol in 20 mM Tris–HCl pH 7.5) and incubated at 37°C for one hour, shaking at 1,200 rpm for 5 s every two minutes. Samples were then centrifuged at 21,000 g for 15 min at +4°C and the supernatants removed and saved for analysis. The pellets were washed twice in 500 μl RIPA buffer (50 mM Tris–HCl pH 8.0, 150 mM NaCl, 0.5% Na-deoxycholate, 0.1% SDS, 1% Triton X-100) at RT and dissolved in 20 mM Tris–HCl pH 7.5, 2% SDS, 8M Urea, by sonication (Bioruptor, Diagenode) at maximum setting for 5 min at RT. Samples for SDS–PAGE analysis were mixed with 4X LDS Loading Buffer (Life Technologies) supplemented with DTT (Sigma) to 100 mM final concentration and heated for 10 min at 70°C before being loaded on SDS–PAGE gels (Life Technologies).

## Video recording of RNase-treated cell lysate

Jurkat T-cell lysate at 10 mg/ml, prepared as described above, was divided into two quartz cuvettes at RT with or without 5 μl RNase A/T1 or Vehicle and mixed. Recording was done with a Canon

digital camera and started immediately (time 0) taking 60 frames/ second for a total of 30 min.

## Immobilisation of RNase A

100 μg RNase A at 1 μg/μl was coupled to Tosyl activated magnetic beads (Life Technologies) for 20 h at 37°C according to the manufacturer's instructions. After quenching and washing, the coupled RNase A was re-suspended in 0.1% BSA in PBS and kept at +4°C until use. Approximately 50% activity remained after coupling, as determined on yeast tRNA using the RiboGreen kit (Life Technologies).

## Inhibition of RNase A and addition of pre-hydrolysed RNA

RNase A inhibition: 200 μg lysate was mixed with 0.1 μl RNase A (~3 mg/ml) and increasing concentrations of RNasin (Promega), as indicated. Hydrolysis of RNA: 40 μg of total RNA in TE buffer (10 mM Tris–HCl pH 8.0, 1 mM EDTA) was incubated with 10 μl immobilised RNase A for 1 h at 37°C. RNase A was removed by magnetic separation and the hydrolysed RNA mixed with 120 U RNasin and kept on ice until used. Alternatively, 40 μg total RNA in 0.1 M NaOH was incubated at 85°C for 1 h and then adjusted to pH 7.5 with 1 M Tris–HCl pH 7.0. RNase A digested and NaOH hydrolysed RNA was then added to 200 μg of neuronal lysate, prepared as outlined above, and incubated at 37°C for 1 h. Aggregated and soluble proteins were collected as before and analysed by SDS–PAGE.

## Cloning and use of TDP-43 and Aβ

All PCRs were performed with Q5 polymerase (NEB) according to the manufacturer's instructions. Human Aβ 1-40 was PCR-amplified from full length APP (Origen, #RC209575) using Abeta_XhoI_F and Abeta_BamHI_R primers (Table EV1), purified and then cleaved with Xho I and Bam HI (both NEB). After further purification, the fragments were ligated into the Xho I and Bam HI sites of pEGFP-N3 (Clontech), creating Aβ fused in frame to the N-terminus of GFP. HEK293T cells, plated at a density of 0.2 × 10$^6$ cells/well in a 24-well plate, were transfected with Aβ-GFP or empty vector using FuGENE HD (Promega). For each well, we used 0.6 μg DNA and 2 μl FuGENE HD in a total volume of 30 μl OptiMEM (Life Technologies). Cells were harvested 48 hours after transfection and washed in PBS and then either stored at −80°C or used directly. Thawed or fresh cells were lysed in 80 μl Lysis Buffer 1 as described above and treated with RNase A/T1 for one hour at 37°C. Aggregated proteins were collected by centrifugation and samples processed and analysed by SDS–PAGE as described above.

Human TDP-43 was PCR-amplified with TARDBP BspHI and TARDBP Not I primers (Table EV1) from cDNA, prepared from Jurkat RNA using Superscript II (Thermo Fisher Scientific) according to the manufacturer's instructions. Purified product was cleaved with Nco I and Not I and cloned into pA4D5-8mRFP (Markiv *et al*, 2011), creating TDP-43 with a C-terminal His tag. Ligated plasmids were transformed into BL21 (DE3) cells (C2527, NEB) and correct clones verified by Sanger sequencing. A single colony was grown overnight at 37°C in Luria-Bertani Broth (LB) in the presence of 2.5% glucose and ampicillin (50 μg/ml). We found that the presence of glucose, which further suppresses the T7 promoter in

pA4D5-8mRFP, was critical to allow the expression of TDP-43. The overnight culture was diluted 20 times in fresh LB medium with glucose (2.5%) and ampicillin (50 μg/ml) and grown at 37°C until an $OD_{600}$ of approximately 0.8. The bacteria were then pelleted by centrifugation and re-suspended in fresh LB supplemented with 1 mM IPTG (GEN-S-02122-5, Generon) to induce expression, and left shaking at 250 rpm at RT for 2 hours. Recombinant TDP-43 was purified using the Ni-NTA Spin Column purification kit (31014, Qiagen), essentially according to the manufacturer's instructions. Briefly, bacteria were lysed in Buffer 1 (6 M GuHCl, 0.1 M Na Phosphate, 10 mM Tris–HCl pH 8.0) by sonication and insoluble debris cleared by centrifugation (21,000 *g*, 30 min). Spin column-captured proteins were washed twice in Buffer 1 adjusted to pH 6.3 and thrice in Buffer 1 adjusted to pH 4.5 and then eluted in 6 M GuHCl, pH 2.0. Purity was determined by separating a TCA-precipitated (to remove GuHCl) sample on an SDS–PAGE gel followed by analysis on a Bioanalyzer Protein-230 chip and was judged to be more than 80% (Appendix Fig S1). 5 μg recombinant TDP-43 was used for renaturation, in the presence or absence of various amounts of total Jurkat RNA, using the procedure for "renaturation through dialysis" described below. To quench released $Ni^{2+}$ in the protein samples before renaturation, EDTA was added to a final concentration of 10 mM before the addition of DTT (see procedure below).

### Nucleic acid-mediated renaturation by dialysis

Proteins were isolated from neuronal or Jurkat cell lysate by RNase A/T1 treatment and centrifugation. Pelleted proteins were dissolved in 50 μl of denaturation buffer (20 mM Tris–HCl pH 7.5, 6 M Guanidine hydrochloride, 1% Triton X-100, 20 mM DTT) and sonicated for 5 min at RT. The protein concentration was determined with the BCA kit (Thermo Fisher) and diluted to 0.4 μg/μl in denaturation buffer. 30–100 μg of solubilised proteins was mixed with 0.5X, in μg, of RNA, DNA or heparin (all in TE buffer). All nucleic acids were heat-denatured at 96°C for 3 min and then rapidly cooled on ice before addition to the denatured proteins. The nucleic acid/ protein mixture was transferred to dialysis tubes (see below) equipped with a 6–8.000 kDa cut-off membrane (Spectrum Lab). Dialysis was performed against 600 ml PBS buffer at 4°C overnight, after which the PBS was replaced with fresh PBS (400 ml) and the container placed in a water bath and kept at 37°C for 1 h. The dialysed samples were transferred to 1.5-ml tubes and the volume adjusted to 100-200 μl with PBS. 7.5–10% of this was taken as Input. Aggregated proteins (Pel 1) were pelleted by centrifugation at 21,000 *g* for 10 min at +4°C, washed twice in RIPA buffer and processed for SDS–PAGE as before. 7.5–10% of the supernatant was saved (Sup 1) and the remaining supernatant was either divided into two new tubes supplemented with 0.5 μl vehicle or 0.5 μl RNase A/T1 or the whole sample placed in one tube and treated with 0.5 μl RNase A/T1. All samples were incubated at 37°C for one hour and centrifuged as before. Pelleted proteins (Pel 2) were washed as before and dissolved in SDS/Urea and sonicated. Equal volumes of each fraction were separated on SDS–PAGE gels and then either stained with coomassie or transferred to membranes for Western blot analysis.

Dialysis tubes were prepared by drilling a 3-mm hole in the lid of a 1.5-ml microcentrifuge tube (Crystal Clear, StarLab). The tube was then cut 1 cm from the top and a new intact lid inserted at the

bottom. After sample addition, the tube was sealed with a dialysis membrane and capped with the drilled lid. This creates a dialysis tube where one end is in contact with the surrounding solution, separated by the membrane. Tubes were placed in the dialysis solution with the holed side facing down.

### Nucleic acid-mediated renaturation through two-step dilution

Proteins aggregated by enzymatic RNA degradation were isolated and denatured in denaturation buffer (20 mM Tris–HCl pH 7.5, 6 M Guanidine hydrochloride, 1% Triton X-100, 20 mM DTT) as described above. The protein concentration was determined with the BCA kit (Thermo Fisher) and diluted to 5.55 μg/μl in denaturation buffer. Typically, 30 μg of denatured proteins, on ice, was mixed (by vortexing) with indicated amounts of nucleic acids, prepared in 1 X TE buffer, to achieve a 1:10 dilution, e.g. 5 μl of denatured proteins + 45 μl of diluted nucleic acids. All nucleic acids were heat-denatured (96°C, 3 min) and cooled on ice before use, except when pre-annealed oligonucleotides were used. Pre-annealed oligonucleotides were prepared in TE buffer supplemented with 100 mM NaCl and placed in a PCR machine at 96°C for 3 min followed by cooling to RT for approximately 45 min. All protein/ nucleic acid mixtures were incubated for 5 min on ice and then diluted 10× with cold Renaturing buffer (10 mM Tris–HCl pH 7.4, 30 mM NaCl), e.g. 50 μl original mixture + 450 μl of Renaturing buffer. After incubation for 10 min on ice, samples were placed at 37°C for 1 h, shaking at 1,200 rpm for 3 s every second minute. Aggregated and soluble proteins were separated by centrifugation, 21,000 *g* for 30 min. Aggregated proteins were solubilised in 20 mM Tris–HCl pH 7.5, 2% SDS, 8M Urea by sonication (Bioruptor, Diagenode) at maximum setting for 5 min at RT and either used for SDS–PAGE or the amount of protein determined using the BCA assay. When determining the amount of protein, the whole pellet was used in a typical 225 μl reaction, e.g. 25 μl sample + 200 μl BCA reagent. In the figures, the amount of aggregation is usually expressed as a percentage of the amount of aggregation observed in the vehicle control, i.e. in samples treated with TE buffer only.

### SDS–PAGE and Western blot analysis

Heated samples were separated on 4–12% Bis-Tris gels (Life Technologies) in MOPS or MES buffer and either transferred to 0.2 μm nitrocellulose or 0.45 μm PVDF membranes (both GE Healthcare) for 2 h at 45V on ice or, alternatively, used directly for coomassie staining (ProtoBlue, National Diagnostics) according to the manufacturer's protocol. In figures with Input and Supernatants, these represent 10%, typically 30 μg of protein, while for the Pellets the full amount was loaded. After transfer, membranes for Western blot were blocked for one hour at RT in 5% milk in TBS-T (50 mM Tris–HCl pH 7.5, 150 mM NaCl, 0.05% Tween-20) and incubated with primary antibodies in the same solution or TBS-T/5% BSA overnight at +4°C. Membranes were then washed 4 × 5 min in TBS-T and incubated for 1 h at RT with HRP-conjugated secondary antibodies diluted in 5% milk/TBS-T. Membranes were then washed as before and incubated for 5 min in ECL Prime (GE Healthcare) before being exposed to films (Thermo Fisher). Primary antibodies used were: TDP-43 (NEB, #G400), HTT (NEB, #D7F7), FUS (Santa Cruz, #sc-47711), MAPT (NEB, #Tau46), NF-H (mouse, Covance, #SMI-

32R), NF-H (rabbit, Sigma, # N4142), Aβ 6E10 (Covance, #SIG-39320), ACTB (Sigma, #A2228), RPL7 (Abcam, #ab72550), PABP (Abcam, #ab21060), GFP (Abcam, #ab1218), HNRNPA1 (Protein-Tech, #11176-1-AP) and HNRNPD (ProteinTech, #12770-1-AP). All primary antibodies were used at 1:1,000 dilution, except ACTB (1:4,000), NF-H (1:4,000), GFP (1:2,000), FUS (1:100) and RPL7 (1:2,000). As secondary antibodies, we used Donkey anti-Rabbit HRP (#NA934V) or Sheep anti-Mouse HRP (#NXA931), both from GE Healthcare, diluted 1:50,000 in 5% milk-TBS-T.

### Isolation and renaturation of brain protein aggregates

Brain tissues (~200 mg frozen) were homogenised for approximately 30 s in 2 ml 0.8 M NaCl, 1% Triton X-100, 0.1 M EDTA, 0.01 M Tris–HCl pH 7.4, 1 mM DTT using a TissueRuptor (Qiagen). The homogenate was then centrifuged for 5 min at 2,500 $g$ at 4°C and the supernatant transferred to a new tube on ice followed by further homogenisation using a 27-G needle and syringe. The samples were then centrifuged at 21,000 $g$ at 4°C for 30 min and the supernatant collected. SDS and Na-deoxycholate were added to the supernatant to 0.1 and 0.5% final concentrations, respectively, and incubated for 10 min at RT. 500 μl of each sample was overlaid on 800 μl of sucrose cushion (1 M sucrose, 0.8 M NaCl, 1% TX-100, 0.5% Na-deoxycholate, 0.2% SDS, 50 mM Tris–HCl pH 7.8) and centrifuged at 167,000 $g$ for 2 h at 4°C. The supernatants were decanted and discarded and the pellet re-suspended in PBS supplemented with NaCl to 500 mM and transferred to new tubes. Pellets were washed twice in PBS/0.5 M NaCl by centrifugation at 21,000 $g$ for 30 min and then dissolved in 6 M GuHCl, 1% Triton X-100 and 20 mM Tris–HCl by sonication. Brain and Jurkat samples were diluted to 2 μg/μl, and 30 μg used for renaturation through dilution as described above, using total Jurkat RNA or the DNA versions of the M1 × 4 oligonucleotides (Table EV1). Renatured samples were incubated for 1 hour at 37°C, shaking at 1,200 rpm every 5th second. Aggregated proteins (Pellet 1) were collected by centrifugation (21,000 $g$, 1 h, 4°C) and saved for analysis. The supernatants were supplemented with RNase A/T1 and incubated with shaking at 1,200 rpm at 37°C overnight. Aggregated proteins (Pellet 2) were collected by centrifugation as above and the supernatant supplemented with $MgCl_2$, 1 mM final concentration, and Benzonase (sc-391121B, Santa Cruz) and incubated at 37°C for 1 h, shaking as above. Aggregated proteins (Pellet 3) were collected as above. All pellets (1–3) were dissolved in 2% SDS/8M Urea by sonication and analysed by Western blot as described.

### ATP-binding and hydrolysis

100 μg proteins aggregated by enzymatic degradation of RNA in human neuronal or Jurkat cell lysates were renatured with 50 μg of total RNA or Ve (TE buffer) as described above. After dialysis, the samples were adjusted to 250 μl with PBS and centrifuged at 2,000 $g$ for 15 min, to remove protein aggregates that would later co-sediment with the agarose beads. Capturing of ATP-binding proteins was performed on 75 μl of this mixture using 30 μl of Aminophenyl-ATP- or naked-agarose beads (Jena Bioscience) according to the manufacturer's protocol. Elution was performed by two sequential 10-min incubations in 20 μl 1X LDS loading buffer (Life Technologies) supplemented with DTT (100 mM final concentration). One fourth of the eluted samples was separated on 4–12%

NuPage gels (Life Technologies) and the gels stained with coomassie blue (ProtoBlue, National Diagnostics). The remaining eluate from two independent replicates was electrophoresed approximately 1 cm into a 4–12% NuPage gel and the top piece of the gel excised and prepared for mass spectrometry analysis as described below. To measure ATP hydrolysis, we used the ADP-Glo™ Kinase Assay (Promega) according to the manufacturer's instructions. Briefly, 5 μl of renatured proteins were mixed in a white 96-well plate (Santa Cruz Biotechnology) with ATP (100 μM final concentration) and 0.1 μl RNase A/T1 mixture or vehicle (50% Glycerol in 20 mM Tris–HCl pH 7.5), all diluted in 1X PBS, 5 mM $MgCl_2$, 2 mM DTT, in a total volume of 15 μl and incubated at RT for 1.5 h. Non-hydrolysed ATP was removed by the addition of 15 μl of ADP-Glo reagent followed by incubation for 1 h at RT. ADP was converted back to ATP by the addition of 30 μl Kinase Detection Reagent and the emitted light quantified after 1.5 h incubation at RT using a Victor 2 Multilabel plate-reader (Wallac). All samples were run in duplicate and data presented as the mean of three independent replicates.

### RNA and DNA isolation and analysis

RNA and DNA were isolated from cells or cell lysates with TRIzol LS or Isol-RNA Lysis Reagent (Life Technologies and 5 PRIME, respectively), according to the manufacturer's instructions. All RNA and DNA samples was dissolved in either 0.1X (for RNA fragmentation, see below) or 1X TE buffer. RNA was analysed by 1.5% agarose gel electrophoresis and visualised with ethidium bromide.

### Mass spectrometry analysis

30 μg of aggregated proteins in 1 × LDS loading buffer (Life Technologies) supplemented with 100 mM DTT were separated on 4–12% Bis-Tris gels in MOPS running buffer. After coomassie staining, each gel lane was divided into 10 equal gel slices and cut into 1mm cubes. Gel bands were destained and reduced with 5 mM TCEP (Pierce) and alkylated with 50 mM chloroacetamide (Sigma) and then digested with trypsin (Promega) for 16 hours. Samples were desalted using homemade C18 columns and then analysed using a Q Exactive Mass Spectrometer (Thermo) at the Central Proteomics Facility (University of Oxford, UK). Data were analysed using Mascot (Matrix Science) with searches performed against the UniProt Human database. Proteins with a Mascot score greater than or equal to 60 and with two unique peptide sequences were considered to be confidently identified. Keratin hits derived from hair (cuticular) were removed from the list of ATP-precipitated proteins as they likely represent contamination introduced during handling.

### Computational analysis of proteins aggregated by enzymatic RNA degradation

Proteins common to both LC-MS/MS samples were compiled into a list and used for further analysis. Gene ontology analysis was performed with PANTHER (Mi *et al*, 2018) using the proteome of differentiated neural stem cells (Song *et al*, 2019) as reference. The proteins with a relative abundance of more than nine in the reference set (Song *et al*, 2019) were used as background to re-analyse the set of RNase-aggregated proteins using the same tool (PANTHER). Low-complexity regions of 30 or more consecutive

amino acids were identified with SEG (Wootton & Federhen, 1993) using the following parameters: [30 amino acid length][3.2 low-complexity trigger][3.55 high extension complexity]. Unstructured regions were identified with DisEMBL (Linding *et al*, 2003) using the default settings with the following changes: amino acid window of 30, join 2 and threshold 1.75. For statistical evaluation, the results were compared with those obtained by permutation analyses. 1,000 permutations per analysis were performed. The permutations consisted of random sets of proteins (n = 1,603), drawn from the complete set of human proteins (http://www.uniprot.org/downloads, accessed on 07/2013) and analysed using SEG and DisEMBL with the same settings as above. The cumulative distributions of the proportion of low-complexity and unstructured regions were compared with the results obtained from the proteins aggregated by enzymatic degradation of RNA. Statistical analysis for each data set was done using the two-sample Kolmogorov–Smirnov test (Conover, 1999) and corrected for multiple testing using Bonferroni correction.

## Nucleic acid immunoprecipitation and sequencing

### RNA and DNA fragmentation

Nucleic acids were isolated as described above. Total RNA (~400 µg), prepared in 0.1X TE buffer, at 0.5 µg/µl, were chemically fragmented in 50 mM Tris-acetate pH 8.1, 100 mM $CH_3CO_2K$ and 30 mM Mg $(CH_3COO)_2$ by incubating the samples at 96°C for 12 min. The reaction was stopped by transferring the samples to ice and the addition of 0.5 M EDTA to final concentration of 45 mM. Fragmented RNA samples were then mixed with 1/10 volume of Na-acetate (3M, pH 5, Sigma) and 2.5 volumes of EtOH and incubated for 1 h at −80°C. Precipitated RNA was collected by centrifugation: 30 min at 12,000 *g* at +4°C and washed once in 75% EtOH before being air-dried and dissolved in TE. Recovered RNA was spectrophotometrically quantified (NanoDrop, Thermo Fisher Scientific) and kept on ice until used. The average size of fragmented RNA was estimated from an aliquot analysed on a Bioanalyzer chip (RNA9000, Agilent Technologies), according to the manufacturer's instructions. Approximately 70% of the initial amount of RNA was recovered after fragmentation.

Genomic DNA, in TE buffer at 3 µg/µl, was fragmented by several rounds of sonication (Bioruptor, Diagenode) at RT until an average fragment length of approximately 200 bp was achieved.

### Nucleic acid-mediated renaturation

Proteins aggregated by enzymatic degradation of RNA in human neuronal lysates were renatured with fragmented RNA or DNA using the dialysis procedure described above. To achieve efficient renaturation, the amount of fragmented RNA had to be increased from the normal 0.5 times to 1.5 times the amount of protein (weight/weight). After dialysis, samples were transferred to 1.5-ml tubes and incubated at 37°C for 1 h, shaking at 1,200 rpm for 3 s every 1.5 min. Aggregated and soluble proteins were separated by centrifugation (21,000 *g*, 30 min, +4°C). The soluble fraction was collected and divided into several tubes for immunoprecipitation or capturing onto nitrocellulose membranes, see below.

### Native RNA Immunoprecipitation

Human neuronal cell lysates were prepared as described above under "Preparation of cell free lysates from neurons and mouse cortex". 200 µg of lysate was incubated with 3 µg anti-NF-H (N4142,

Sigma), 3 µg anti-Aβ (4G8, SIG-39220, Covance) or 3 µg mouse IgG (Sigma) and incubated rotating for 2 h at +4°C. Antibodies were captured by the addition of 15 µl Protein A magnetic beads (10002D, Thermo Fisher Scientific) pre-blocked in 5% BSA, 30 µg M1 × 4 R oligo in PBS/0.1% Triton X-100 for 1 h at RT and incubated for 30 min at RT. Magnetic beads were collected using a magnet and washed twice in PBS, 0.5% Triton X-100, twice in PBS, 500 mM NaCl, 0.5% Triton X-100, and once in PBS, 1,000 mM NaCl, 0.5% Triton X-100 and once in TE. RNA was isolated with the GeneJET PCR Purification Kit according to the manufacturer's instructions with the following alterations: RNA was eluted from the magnetic beads by the addition of 50 µl 6 M guanidine hydrochloride. The supernatants were collected and mixed with 100 µl Binding Buffer, 100 µl water and 300 µl 2-propanol and loaded onto the column. Co-purified DNA was removed by on-column digestion, using the RNase-Free DNase Set (Qiagen) reagents and protocol. Eluted RNA was then precipitated with Na-Acetate and EtOH, using glycogen (R0561, Thermo Fisher Scientific) as a carrier. DNase-treated samples were dissolved in 10 µl water and used for library preparation using the NEBNext Ultra II Directional RNA Library Prep Kit and Indexing primers (E7760 and E7335, respectively, both NEB) according to the manufacturer's instructions. Libraries were pooled and paired-end sequenced (2 × 75 bp) on the NextSeq 500 platform (Illumina). Two biological replicates of each immunoprecipitation were analysed, except for 4G8 where only one sample was analysed. Data can be accessed at GEO GSE99127.

### RNA Immunoprecipitation of renatured proteins

RNA-renatured proteins (~40 µg) were supplemented with 1 µg anti-NF-H (N4142, Sigma), 1 µg anti-Tau (T9450, Sigma), 1 µg anti-Aβ (4G8, SIG-39220, Covance) or 1 µg anti GFP (11122, Thermo Fisher Scientific) and incubated while rotating for 2 h at +4°C. Antibodies were captured and washed as described above. Samples were eluted with 20 µl 6 M guanidine thiocyanate at RT and RNA-purified from the eluate with Isol (5-PRIME), according to the manufacturer's instructions using 1 µg glycogen (R0561, Thermo Fisher Scientific) as co-precipitant. RNA was dissolved in 30 µg 1× DNAs I buffer and treated with 1 µl DNAase I (2222, Thermo Fisher Scientific) for 30 min at 37°C. The sample volumes were adjusted to 100 µl with TE and extracted with an equal volume of phenol (77617, Sigma) and precipitated with Na-Acetate and EtOH, using glycogen as a carrier. DNase-treated samples were dissolved in 10 µl water and used for sequencing library preparation using the NEBNext Directional Ultra RNA kit and Indexing primers (E7530 and E7335, respectively, both NEB) according to the manufacturer's instructions, except that no initial RNA fragmentation was performed. Libraries were pooled and paired-end sequenced (2 × 75 bp) on the MiSeq platform (Illumina). Two biological replicates of each immuprecipitation were analysed. Data can be accessed at GEO GSE99127.

### DNA-immunoprecipitation and membrane capture of renatured proteins

DNA-renatured samples were prepared as described above and aliquoted for immunoprecipitation or membrane capture. For membrane capture, approximately 40 µg renatured proteins or fragmented DNA alone (20 µg, negative control) were slowly passed through a 0.2 µm nitrocellulose membrane (1060004, GE

Healthcare) pre-blocked with 250 μg Yeast tRNA (Thermo Fisher Scientific). The membranes were then washed three times with 10 mM Tris–HCl pH 7.5, 100 mM KCl, 0.1% Triton X-100; twice with 10 mM Tris–HCl pH 7.5, 500 mM KCl, 0.1% Triton X-100; and once with 1X TE. Captured proteins and nucleic acids were eluted by the addition of 2% SDS/8M Urea and the nucleic acids extracted with Isol (5-PRIME) as described above. Immunoprecipitation with Aβ 4G8 (1 μg) and mouse IgG (1 μg) were performed as described for RNA immunoprecipitation of renatured proteins above using approximately 40 μg of renatured proteins per IP. Protein A magnetic beads were pre-blocked for 1 h at RT with 70 μg Yeast tRNA (Thermo Fisher Scientific). Sequencing libraries were prepared using the NEBNext DNA ultra kit (E7370, NEB), using NEBNext Indexing primers (E7335, NEB), and the libraries pooled and sequenced on the MiSeq system (Illumina) using paired-end reads of 300 bp. Data can be accessed at GEO GSE99127.

### Computational nucleic acid sequence analysis

Reads from both the DNA and RNA-seq experiments were trimmed using Trim Galore with default settings for paired-end reads. RNA reads were aligned to the human reference genome (hg19), using STAR (Dobin *et al*, 2013) using default settings. Potential PCR artefacts in the aligned RNA-seq files were removed by the MarkDuplicates tool in Picard-tools, using default settings. DNA sequences were aligned to the unmasked human reference genome (hg19) using Bowtie2. To identify enriched regions, we employed MACS (Zhang *et al*, 2008), using the GFP (RNA), IgG (Native RNA-IP and DNA-IP) or DNA-only (membrane) samples as negative controls. We used the reads from both biological replicates as input files, except for the Native RNA-IP sample of 4G8, where only one sample passed the quality check. MACS were run with the following settings: band width = 150 for RNA or 188 for DNA, Broad region calling = off (RNA), Searching for subpeak summits = on (RNA) and call-summit = on for DNA. The *annotatePeaks* package in the HOMER software package (Heinz *et al*, 2010), run with default settings, was used to annotate the peak files generated by MACS. For motif discovery, we extracted a 300 nt fragment, centred at the peak detected by MACS. We used the MEME suite of programs (MEME-Chip, (Machanick & Bailey, 2011)) with the following settings: *-meme-mod anr -meme-minw 5 -meme-maxw 50 -meme-nmotifs 6 -dreme-e 0.05 -centrimo-local -centrimo-score 5.0 -centrimo-ethresh 10.0*. Motif similarities were evaluated with Tomtom (in the MEME suite) using default settings.

### Quantification and statistical analysis

Statistical analyses in Figs 3B and E, and 5A, C, D and 6B–D, and EV4C and EV5D were performed by one-way ANOVA followed by *post hoc* analysis with Bonferroni correction for multiple testing using the online tool at: http://astatsa.com/OneWay_Anova_with_TukeyHSD/. Statistical analysis for Figs. EV2D and E was performed using the two-sample Kolmogorov–Smirnov test (Conover, 1999) and corrected for multiple testing using Bonferroni correction. When mentioned, the term "independent samples" refers to biological repeats of the same experiments but using different starting material, e.g. cell lysate.

### Oligonucleotides

All oligonucleotides investigated in this study were purchased from IDT in a desalted form and used without any further purification. Sequences of oligonucleotides are listed in Table EV1.

## Data availability

The DNA and RNA immunoprecipitation data from this publication have been deposited in the Gene Expression Omnibus (GEO) database and assigned the identifier GSE99127 (https://www.ncbi.nlm.nih.gov/geo/query/acc.cgi?acc=GSE99127).

**Expanded View** for this article is available online.

### Acknowledgements

We thank Dr B Thomas and the staff of the Central Proteomics Facility at the University of Oxford (www.proteomics.ox.ac.uk) for mass spectrometry and data analysis services, and Neil Atam for creating the movie. We also thank Prof Michael Curtis, Prof Thomas MacDonald, Dr John Lanham, Dr Michele Hill-Perkins, Dr Pietro Fratta, Mrs. Julie Fennell and Mr. Graham Fennell for their support, and Dr Jarek Szary for discussions. We are grateful to Professor Linda Greensmith for critical review of the manuscript. This work was generously supported by a Strategic Project Grant from Barts Charity; grants from Queen Mary Innovation, The Medical Research Council, The Motor Neurone Disease Association, and Rosetrees Trust; a Wellcome Trust Value in People Award; and a legacy from Mr. Ivan Fennell. In memory of Mr. Ivan Fennell.

### Author contributions

JA conceived and planned the research, performed most of the experiments, analysed the data and wrote the paper. SR, TAJ and RA performed experiments. AM provided ALS samples and strategic planning, evaluated the data and commented on the manuscript. CPC and MRB performed computational analysis of aggregated proteins and nucleic acids. GG supported the study and evaluated the data. DS conceived and planned the research, evaluated the data and wrote the paper together with JA.

### Conflict of interest
The authors declare that they have no conflict of interest.

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
