## [Review Process File · EMBO Reports]

Enzymatic degradation of RNA causes widespread protein aggregation in cell and tissue lysates

Johan Aarum, Claudia Cabrera, Tania Jones, Shiron Rajendran, Rocco Adiutori, Gavin Giovannoni, Michael Barnes, Andrea Malaspina, and Denise Sheer

DOI: [10.15252/embr.201949585](https://doi.org/10.15252/embr.201949585)

Corresponding author(s): Denise Sheer (d.sheer@qmul.ac.uk), Johan Aarum (Johan.aarum@sll.se)

Review Timeline:

Submission Date:	12th Nov 19
Editorial Decision:	10th Jan 20
Revision Received:	27th Jan 20
Editorial Decision:	31st Jan 20
Revision Received:	4th Mar 20
Editorial Decision:	29th May 20
Revision Received:	28th Jun 20
Accepted:	30th Jul 20

Editor: Esther Schnapp

Transaction Report: This manuscript was transferred to EMBO reports following peer review at The EMBO Journal.

9th Jan 2020

Dear Prof. Sheer,

Thank you for the transfer of your revised manuscript to EMBO reports, and also for your patience, given the unusual delay in getting back to you. We had received the comments from referees 1 and 2 in December, but given that they are not in agreement, I contacted a third, new referee. All comments have come in now and are pasted below.

I am sorry to say that the evaluation of your study is not a positive one. As you will see, while referee 2 is positive, referee 1 remains unconvinced that the manuscript reports sufficiently novel concepts. In addition, referee 3 also thinks that more experimental evidence is required to strengthen the findings and render them suitable for publication. Given these concerns and the fact that you already had a chance to significantly revise the manuscript once, I am sorry to say that we cannot offer to publish your manuscript.

That being said, we recognize that your findings will be of value to the field. I would therefore like to propose a transfer of your manuscript and referee reports to the new open-access journal Life Science Alliance. Life Science Alliance is launched as a partnership between EMBO Press, Rockefeller Press, and Cold Spring Harbor Laboratory Press, and publishes work that is of high value to the respective communities across all areas in the life sciences. I have already discussed your work with Andrea Leibfried, executive editor of Life Science Alliance, and Andrea would like to publish your work, pending satisfactory revision. Andrea requests that the remaining referee concerns should be addressed by re-wording. She further suggests that you point out within the manuscript text that only the pellet was used for the MS approach. The ALS part could be removed to address the reviewer's concern.

Andrea will be happy to discuss the process and revision/amendments needed with you at any point. You can also directly discuss the revision with Andrea by contacting her at a.leibfried@life-science-alliance.org

I hope you will view the possibility of a transfer favorably. If this is the case, please use the link below to transfer the manuscript directly.

Referee #1:

In their revised manuscript, Sheer and colleagues attempt to address several of my previous critiques. However, many of my previous points have not been adequately addressed. As noted in my previous review it is not very surprising that bulk removal of RNA elicits widespread protein aggregation as evidenced by many studies in the literature. As such, I do not find this work to be

suitable for EMBO Rep. in terms of novelty or in terms of execution. Many problems remain, which are summarized and reiterated below:

1. It is simply not surprising that RNA-binding proteins aggregate once RNA is eliminated. Moreover, it is not surprising that not all proteins that aggregate once RNA is eliminated are RNA-binding proteins. RNAs are known to be chaperones for proteins as noted in my last review. Moreover, various proteins that engage RNA-binding proteins as well as molecular chaperones are anticipated to coprecipitate with RNA-binding proteins. Hence, the general observation in this manuscript is not surprising. Indeed, for example the Hyman/Alberti labs have shown that direct injection of RNase into the nucleus causes widespread protein aggregation.

2. In the experiment depicted in Fig. 3a, it remains uncertain what types of RNA are contributing to the activity. It remains unclear whether this activity be supported by microRNAs, rRNA, mRNA, long noncoding RNA etc. Since DNA can have the same effect, this would appear to be a nonspecific chemical chaperone effect of nucleic acids above a certain size (as tRNA is ineffective). It remains unclear whether other nucleic acids above a certain size, such as poly(ADP-ribose) or even other long anionic polymers like polyphosphate or heparin would have the same effect. It thus remains completely uncertain whether there is anything special about RNA in this context.

3. In Fig. 3e, the authors attribute stimulated ATPase activity to RNA-mediated refolding. However, in such a crude assay it is not possible to make this conclusion very strongly. It could simply be that RNA stimulates the ATPase activity of a variety of proteins in these crude extracts, which is independent of any effect of refolding, i.e. addition of RNA would stimulate ATPase activity in any crude extract. This possibility is not controlled for and this remains the case despite the rebuttal of the authors. Likewise, it is not clear whether the RNA preparations used here contain any contaminating ATPases. To make this case more strongly, this protocol (i.e. renaturation in the presence of RNA) should be employed on single, purified ATPases (such as any of the ATPases that precipitate upon RNA depletion, e.g. VCP) where the appropriate controls can be readily performed. These issues remain unaddressed and remain unanswered in the revised manuscript.

4. The authors focus on the properties of nucleic acids that help maintain the solubility of previously aggregated protein after they have been dissolved with 6M GuHCl. However, it is unclear whether there is anything special about RNA (see point 2 above). This activity is very different from nucleic acids directly resolubilizing protein aggregates as the authors appear to suggest throughout the paper (despite changes to the abstract). The nucleic acids depend on 6M GuHCl to first actually solubilize the aggregates, and then merely chaperone (as opposed to disaggregate) the proteins as the GuHCl is removed. This important point is still not evident throughout the text where the authors overstate their findings.

5. The application of their protocol to ALS patient samples is not informative as they focus on NF-H, which is of no relevance to the disease. They could have taken this approach with any brain lysate and so it is misleading to suggest that their approach has any relevance to ALS. It continues to remain unclear why they did not assess TDP-43 or other ALS-relevant proteins (e.g. c9-DPRs, FUS, SOD1), which in some cases are extremely abundant proteins.

6. They continue to ignore several key papers from the literature perhaps to make their work seem more novel. For example, several excellent papers are not cited including: Suljoadikusumo et al., 2001 and Sun et al., 2014.

7. The authors still do not seem to understand the difference between protein aggregation and

liquid-liquid phase separation.

References:

Suljoadikusumo, I., Horikoshi, N., and Usheva, A. (2001). Another function for the mitochondrial ribosomal RNA: protein folding. *Biochemistry* 40, 11559-11564.

Sun, Y., Arslan, P.E., Won, A., Yip, C.M., and Chakrabarty, A. (2014). Binding of TDP-43 to the 3'UTR of its cognate mRNA enhances its solubility. *Biochemistry* 53, 5885-5894.

Referee #2:

The authors have addressed all the concerns and the paper should be published.

Referee #3:

Aarum et al reports here an interesting studies showing that RNA degradation causes precipitation of proteins. I've read the manuscript and the provided reviews from referee 1 and 2. See my points below:

1. Which proportion of the protein precipitates. I searched for information across the text that could provide any clue about the penetrance of the effect but this info is not available (or I couldn't find it). Which % of the total protein precipitates? This should be disclosed in all figures in the text to differentiate between a real major effect and a marginal consequence of loss of RNA.

2. Looking at the silver staining in figure 1, I don't see major differences in profile from the input and the pellet. This would be expected if the precipitation would be biased toward a given pool of proteins, e.g. RBPs (see RBP silver staining pattern in Baltz et al. *Mol Cell* 2012 and Castello et al *Cell* 2012). This is supported by the precipitation of ACTB, which is an abundant protein that, to my knowledge, does not interact with RNA.

3. The proteomic analysis is biased. Authors only analysed (to my knowledge) the pellet. If you mass spec a whole cell lysate in a shotgun approach, near 30% of the identified proteins (normally from 3000 to 5000) will be RBPs just because many of the housekeeping RBPs (ribosomal protein, hnRNPs, etc) are very abundant. For this experiment to be informative it is necessary to analyse the input and the supernatant and perform the enrichment analyses against these datasets. By doing so, it would be possible to correct these biases and provide new cues about the proteins within the whole cell proteome that exhibit a differential behaviour in absence of RNA. If authors are not able to perform such experiments, at least, should use a whole proteome similar in size and generation conditions in a suitable line. There are a few made in HEK293 and HeLa.

4. I agree with referee 2 that authors should highlight better in what points this work differ from previous related works. I see a strong point is the proteomics, however, see my points in comment 3.

** As a service to authors, EMBO Press provides authors with the ability to transfer a manuscript that one journal cannot offer to publish to another journal, without the author having to upload the manuscript data again. To transfer your manuscript to another EMBO Press journal using this service, please click on Link Not Available

Response to Referees: Aarum et al, EMBOR-2019-49585V1, 24th January 2020

We thank the Referees for their reviews of our manuscript.

Referee #1:

In their revised manuscript, Sheer and colleagues attempt to address several of my previous critiques. However, many of my previous points have not been adequately addressed. As noted in my previous review it is not very surprising that bulk removal of RNA elicits widespread protein aggregation as evidenced by many studies in the literature. As such, I do not find this work to be suitable for EMBO Rep. in terms of novelty or in terms of execution. Many problems remain, which are summarized and reiterated below:

The arguments of this Referee are difficult to understand as, on the one hand he/she argues that our findings, i.e. the effect of RNA on protein solubility, are well established and obvious (point 1 below), but on the other hand questions whether our findings have anything to do with RNA in the first place (point 2, 3 and 4). In addition, the reference to the Hyman/Alberti paper is confusing as it describes how liquid-liquid phase of recombinant FUS and TDP-43 are affected when RNase is injected into cells (no widespread aggregation shown) while at the same time the Referee appears in point 7 to suggest that liquid-liquid phase separation is unrelated to protein aggregation.

1. It is simply not surprising that RNA-binding proteins aggregate once RNA is eliminated. Moreover, it is not surprising that not all proteins that aggregate once RNA is eliminated are RNA-binding proteins. RNAs are known to be chaperones for proteins as noted in my last review. Moreover, various proteins that engage RNA-binding proteins as well as molecular chaperones are anticipated to coprecipitate with RNA-binding proteins. Hence, the general observation in this manuscript is not surprising. Indeed, for example the Hyman/Alberti labs have shown that direct injection of RNase into the nucleus causes widespread protein aggregation.

In the paper we discuss the possibility of co-aggregation, page 7 last paragraph and in the Discussion p13-14, and have also added experiments on recombinant TDP-43 to further investigate this. Also, and fundamentally, one of the primary reasons we perform the denaturation/renaturation experiments is to investigate if what we observe upon RNase-treatment is the effect of co-sequestering, i.e. that a few RNA-binding proteins cause the precipitation of other, unrelated proteins. If this would be the case, refolding with RNA would mainly affect these RNA-binding proteins while the other, co-sequestered proteins (“various proteins that engage RNA-binding proteins as well as molecular chaperones”) would be expected to remain insoluble. This is simply not the case as most, if not all, of the proteins refolded in the presence of RNA are soluble – but aggregate again if RNA is removed, see for example Figs 3, 5 and 6.

The Maharana paper from the Hyman/Alberti labs is included and discussed in several places in our manuscript.

We reiterate our argument that while previous studies have described the effect of RNA on single or a few RNA-binding proteins, we suggest that RNA actually has a much wider activity in modulating protein solubility. These findings are important as they have conceptual and practical implications for several biological fields, as stated before.

2. In the experiment depicted in Fig. 3a, it remains uncertain what types of RNA are contributing to the activity. It remains unclear whether this activity be supported by microRNAs, rRNA, mRNA, long noncoding RNA etc. Since DNA can have the same effect, this would appear to be a nonspecific chemical chaperone effect of nucleic acids above a certain size (as tRNA is ineffective). It remains unclear whether other nucleic acids above a certain size, such as poly(ADP-ribose) or even other long anionic polymers like polyphosphate or heparin would have the same effect. It thus remains completely uncertain whether there is anything special about RNA in this context.

Fig. 3a outlines the principle of the experiments used in Fig 3 and other experiments which follow. Addition of DNase has no effect on protein aggregation in cell lysate, but RNase does, strongly arguing that RNA is important for maintaining protein solubility in the lysate. Regarding other polyanions, we have performed the same experiment as with RNA and DNA using heparin (page 7, second last paragraph and Expanded View Fig. 4a). In this experiment heparin has no effect.

We then use the same assay with short defined DNA oligonucleotides (60-90 nts, similar in size to tRNA) that differ in their sequence and theoretical structure. Here it is clear that oligonucleotides of similar composition (same proportion of purines and pyrimidines) but with different structural configurations have very different capacities to promote the refolding of aggregated proteins (Figures 5 and 6). If this was a nonspecific chemical chaperone effect, one would expect no difference between these oligonucleotides but a difference is obvious.

Based on our RNA sequencing data several types of RNA can maintain protein solubility after denaturation and refolding, including ncRNA and mRNA.

3. In Fig. 3e, the authors attribute stimulated ATPase activity to RNA-mediated refolding. However, in such a crude assay it is not possible to make this conclusion very strongly. It could simply be that RNA stimulates the ATPase activity of a variety of proteins in these crude extracts, which is independent of any effect of refolding, i.e. addition of RNA would stimulate ATPase activity in any crude extract. This possibility is not controlled for and this remains the case despite the rebuttal of the authors. Likewise, it is not clear whether the RNA preparations used here contain any contaminating ATPases. To make this case more strongly, this protocol (i.e. renaturation in the presence of RNA) should be employed on single, purified ATPases (such as any of the ATPases that precipitate upon RNA depletion, e.g. VCP) where the appropriate controls can be readily performed. These issues remain unaddressed and remain unanswered in the revised manuscript.

We do not agree with the referee on this point. Firstly, if the RNA had no effect on the solubility of the proteins, then there should be ATPase activity in the samples without RNA – but this is not the case (Fig 3e and Expanded view Fig 4d). Secondly, our statement that RNA helps to refold ATPases to a functional state should also be viewed in light of the binding of ATPases to ATP-agarose (Fig 4a and Expanded View Spreadsheet 2) which only occurs if the proteins are refolded in the presence of RNA.

4. The authors focus on the properties of nucleic acids that help maintain the solubility of previously aggregated protein after they have been dissolved with 6M GuHCl. However, it is unclear whether there is anything special about RNA (see point 2 above). This activity is very different from nucleic acids directly resolubilizing protein aggregates as the authors appear to suggest throughout the

paper (despite changes to the abstract). The nucleic acids depend on 6M GuHCl to first actually solubilize the aggregates, and then merely chaperone (as opposed to disaggregate) the proteins as the GuHCl is removed. This important point is still not evident throughout the text where the authors overstate their findings.

We state on page 7, second paragraph and in Figure 3a that the proteins are first denatured in GuHCl followed by renaturing with different additives, e.g. RNA, oligonucleotides etc. If this needs further clarification for each of the following experiments, we are happy to do this. We could also change the text to be more clear, e.g. page 9, second paragraph: From “First, to identify DNA sequences capable of renaturing aggregated proteins...” to “First, to identify DNA sequences capable of renaturing GuHCl-denatured proteins...”

5. The application of their protocol to ALS patient samples is not informative as they focus on Nf-H, which is of no relevance to the disease. They could have taken this approach with any brain lysate and so it is misleading to suggest that their approach has any relevance to ALS. It continues to remain unclear why they did not assess TDP-43 or other ALS-relevant proteins (e.g. c9-DPRs, FUS, SOD1), which in some cases are extremely abundant proteins.

Our reference to and work on Nf-H and on ALS serve completely different purposes, and we have not suggested that Nf-H is a key pathogenic protein in ALS as are proteins encoded by genes involved in familial ALS (TDP-43; C9orf72 etc). The choice of Nf-H is dictated by the simple fact that it is a protein which is frequently encountered in brain aggregates, particularly in the neurodegenerative process seen in ALS and also in circulation. The choice of ALS as a paradigm of neurodegeneration is because it is a condition characterised by the formation of protein aggregates.

Furthermore, we disagree with the statement “ NfH has no relevance to the disease”. While as discussed above it is clear that NfH is not the product of a causative genetic mutation in ALS, it has been heavily implicated in pathogenesis of the disease. Evidence includes the observation of abnormal NfH subunit accumulation in neuronal perikarya and spheroids in affected spinal cord areas, the risk of developing ALS associated with polymorphisms of the NfH gene and, more importantly, NfH being one the most informative neurochemical markers of the disease whereby its fluid expression has strong specificity and sensitivity in the diagnostic definition of ALS and is linked to rate of progression.

Two references are given below as examples, but there are many others in the literature:

*Lu CH, Petzold A, Topping J, Allen K, Macdonald-Wallis C, Clarke J, Pearce N, Kuhle J, Giovannoni G, Fratta P, Sidle K, Fish M, Orrell R, Howard R, Greensmith L, Malaspina A. Plasma neurofilament heavy chain levels and disease progression in amyotrophic lateral sclerosis: insights from a longitudinal study. *J Neurol Neurosurg Psychiatry*. 2015 May;86(5):565-73. doi: 10.1136/jnnp-2014-307672.*

*Xu Z, Henderson RD, David M, McCombe PA. Neurofilaments as Biomarkers for Amyotrophic Lateral Sclerosis: A Systematic Review and Meta-Analysis. *PLoS One*. 2016 Oct 12;11(10):e0164625. doi: 10.1371/journal.pone.0164625.*

6. They continue to ignore several key papers from the literature perhaps to make their work seem more novel. For example, several excellent papers are not cited including: Sulijoadikusumo et al., 2001 and Sun et al., 2014.

We are surprised by the extraordinary allegation that we are deliberately ignoring key papers so that our work seems more novel. This comment indicates a prejudicial bias in this referee's report. We included most but not all the references previously suggested by the referee, simply in the interest of space and where there was overlap. We would be happy to include the Sulijoadikusumo and Sun references in the final version.

7. The authors still do not seem to understand the difference between protein aggregation and liquid-liquid phase separation.

We believe that this refers to the Introduction where we touch upon how phase-separation and protein aggregation could be linked. There are several excellent reports and reviews substantiating such a connection, for example Lin et al., 2015 and Elbaum-Garfinkle 2019. We would therefore argue that this information is relevant in its current form.

References:

Lin et al., Formation and Maturation of Phase-Separated Liquid Droplets by RNA-Binding Proteins. Molecular Cell, 2015, 60: 189-192

Elbaum-Garfinkle, Matter over mind: Liquid phase separation and neurodegeneration. J Biol Chem. 2019 3;294(18):7160-7168

Referee #2:

The authors have addressed all the concerns and the paper should be published.

Referee #3:

Aarum et al reports here an interesting studies showing that RNA degradation causes precipitation of proteins. I've read the manuscript and the provided reviews from referee 1 and 2. See my points below:

1. Which proportion of the protein precipitates. I searched for information across the text that could provide any clue about the penetrance of the effect but this info is not available (or I couldn't find it). Which % of the total protein precipitates? This should be disclosed in all figures in the text to differentiate between a real major effect and a marginal consequence of loss of RNA.

In a typical experiment, 10-15% of the total amount of soluble proteins in the lysate are precipitated upon RNase treatment. This information can easily be added to relevant figures.

2. Looking at the silver staining in figure 1, I don't see major differences in profile from the input and the pellet. This would be expected if the precipitation would be biased toward a given pool of proteins, e.g. RBPs (see RBP silver staining pattern in Baltz et al. Mol Cell 2012 and Castello et al Cell 2012). This is supported by the precipitation of ACTB, which is an abundant protein that, to my knowledge, does not interact with RNA.

We appreciate that this is not very clear and believe this is due to the relatively large amount, both in terms of mass and numbers, of proteins present in the pellet and supernatant together with the resolution of the Coomassie gel where proteins of similar size and amount will be difficult to distinguish between lanes. Similar results have been obtained by others using a comparable approach (b-isox which precipitates proteins with low complexity regions, Kato et al., Cell. 2012, 149(4): 753). This can be clarified in the text.

3. The proteomic analysis is biased. Authors only analysed (to my knowledge) the pellet. If you mass spec a whole cell lysate in a shotgun approach, near 30% of the identified proteins (normally from 3000 to 5000) will be RBPs just because many of the housekeeping RBPs (ribosomal protein, hnRNPs, etc) are very abundant. For this experiment to be informative it is necessary to analyse the input and the supernatant and perform the enrichment analyses against these datasets. By doing so, it would be possible to correct these biases and provide new cues about the proteins within the whole cell proteome that exhibit a differential behaviour in absence of RNA. If authors are not able to perform such experiments, at least, should use a whole proteome similar in size and generation conditions in a suitable line. There are a few made in HEK293 and HeLa.

We have not analysed the input or the supernatant. When we do the enrichment studies, we use the whole theoretical human proteome but agree that a better comparison would be to a factual, cell-type specific dataset. This can be done.

4. I agree with referee 2 that authors should highlight better in what points this work differ from previous related works. I see a strong point is the proteomics, however, see my points in comment 3.

We can do this.

Dear Prof. Sheer,

Thank you for your email asking us to reconsider our decision on your manuscript. I have discussed your point-by-point response with my colleagues now, and I am sorry to say that we have decided to remain by our decision that we cannot offer to publish your revised manuscript as it stands now.

However, I would be willing to consider a newly revised manuscript that addresses all referee concerns. If at least 2 referees will support the publication of your work, we can offer to publish it. Please submit the newly revised manuscript as a new submission and mention in the cover letter the manuscript history here at our journal. Please also submit a detailed point by point response with the revised manuscript, and I will try to secure the same referees for it.

I hope that you agree that this is a good way forward.

** As a service to authors, EMBO Press provides authors with the ability to transfer a manuscript that one journal cannot offer to publish to another journal, without the author having to upload the manuscript data again. To transfer your manuscript to another EMBO Press journal using this service, please click on

Link Not Available

Dr Esther Schnapp
Senior Editor
EMBO Reports
Meyerhofstrasse 1
D-69117 Heidelberg
Germany

3rd March 2020

Dear Dr Schnapp

New submission: Aarum et al, Enzymatic degradation of RNA causes widespread protein aggregation in cell and tissue lysates

Thank you once again for your kind consideration of our previous submission.

We have conducted additional analyses and have revised the manuscript in response to the referees' and your comments, and would like to make a new submission to EMBO Reports.

Protein aggregation, in its most general sense, is part of normal physiology but it is also the unifying theme of several human disorders, in particular neurodegenerative diseases. For example, A β and tau aggregates are the hallmarks of Alzheimer's disease while α -synuclein characterises Parkinson's disease. Intriguingly, the same protein can be found aggregated in several diseases: α -synuclein is found aggregated in both Alzheimer's disease and Parkinson's disease, while aggregated actin, in the form of Hirano bodies, is found across all neurodegenerative diseases. This raises the possibility that a common mechanism controls the non-aggregated state of these proteins.

The interplay between RNA and certain proteins, in particular RNA-binding proteins, has recently been shown to affect protein aggregation (Kovachev et al, Scientific Reports, 2019). In these studies, high ratios of RNA to proteins promotes solubility while low ratios lead to aggregation. These effects are reported to be largely independent of nucleic acid structure or sequence.

In our manuscript, we show that these RNA-protein associations underpin a more general mechanism that affects protein aggregation. We show that many proteins (>1,300), the majority without any known association to RNA and including several linked to disease, require RNA for their solubility and, at least for some, their functionality. In contrast to previous studies, we show that this phenomenon is strictly dependent on nucleic acid structure, with synthetic DNA oligonucleotides of similar compositions but different configurations having very different effects. In a crucial link to disease, we also show that protein aggregates isolated from patients with Amyotrophic Lateral Sclerosis can efficiently be maintained in a soluble state after refolding with nucleic acids, pointing to potential future therapeutic interventions.

Our findings have important implications as they indicate a fundamental mechanism that influences protein solubility and, likely, protein folding. In addition, given the dependence on nucleic acid structure, our report sheds light on some of the observed contradictory effects of nucleic acids and polyanions on protein aggregation, and fits well with the current interest in protein phase-separation, e.g. Maharana et al, Science, 2018, and the emerging realisation of the importance of nucleic acids for protein aggregation. Our study should thus stimulate further investigations across several disciplines, ranging from mechanisms of disease to basic cell biology and biochemistry.

- / cont.

There are also practical consequences of our findings, as they may influence how we analyse and treat biological solutions. For example, cell lysate and protein solutions left on the bench or defrosted develop protein precipitates. As these observations can now easily be explained in terms of altered RNA-protein associations, strategies to prevent aggregation can be developed. Indeed, in our manuscript we show that simply by adding a synthetic DNA oligonucleotide to a cell lysate we can completely prevent this from happening. Similarly, our renaturation method using nucleic acids vastly outperforms current technologies – what often takes days with poor yield can now be achieved in minutes with extremely high efficiency. This has obvious ramifications for situations where large quantities of renatured proteins are required, e.g. for structural and mechanistic studies, but also for the biopharmaceutical industry, both of which are plagued by unwanted protein aggregation.

We have included in our submission our detailed Response to the Referees' Reports, showing full details of where the changes have been made in the new manuscript.

We understand that our colleague in the USA, Dr Scott Horowitz, has resubmitted a revised manuscript to EMBO Reports: Begeman et al, G-Quadruplexes Act as Sequence Dependent Chaperones via Protein Oligomerization. As their findings on nucleic acids acting as chaperones for folding of certain proteins are closely related to ours (using a completely different experimental system and methodology), please could you consider co-publication of our two papers.

We are looking forward to hearing from you.

Thank you.

Yours sincerely,

Johan Aarum, PhD.

Denise Sheer, D.Phil.

Response to Referees: Aarum et al, EMBOR-2019-49585V1

We thank the referees for their detailed and helpful reviews of our manuscript. We have conducted additional analyses and revised our manuscript as they suggested, and have addressed each of their comments below.

Referee #1:

In their revised manuscript, Sheer and colleagues attempt to address several of my previous critiques. However, many of my previous points have not been adequately addressed. As noted in my previous review it is not very surprising that bulk removal of RNA elicits widespread protein aggregation as evidenced by many studies in the literature. As such, I do not find this work to be suitable for EMBO Rep. in terms of novelty or in terms of execution. Many problems remain, which are summarized and reiterated below:

The arguments are difficult to understand as, on the one hand the referee argues that our findings, i.e. the effect of RNA on protein solubility, are well established and obvious (point 1 below), but on the other hand questions whether our findings have anything to do with RNA in the first place (point 2, 3 and 4). In addition, the reference to the Hyman/Alberti paper is confusing as it describes how liquid-liquid phase of recombinant FUS and TDP-43 are affected when RNase is injected into cells (no widespread aggregation shown) while at the same time the referee appears in point 7 to suggest that liquid-liquid phase separation is unrelated to protein aggregation.

1. It is simply not surprising that RNA-binding proteins aggregate once RNA is eliminated. Moreover, it is not surprising that not all proteins that aggregate once RNA is eliminated are RNA-binding proteins. RNAs are known to be chaperones for proteins as noted in my last review. Moreover, various proteins that engage RNA-binding proteins as well as molecular chaperones are anticipated to coprecipitate with RNA-binding proteins. Hence, the general observation in this manuscript is not surprising. Indeed, for example the Hyman/Alberti labs have shown that direct injection of RNase into the nucleus causes widespread protein aggregation.

In the paper we discuss the possibility of co-aggregation, Results page 8, third paragraph describing our new experiments on recombinant TDP-43 to investigate this further, and in the Discussion page 14, second paragraph. Also, and fundamentally, one of the primary reasons we perform the denaturation/renaturation experiments is to investigate if what we observe upon RNase-treatment is the effect of co-sequestering, i.e. that a few RNA-binding proteins cause the precipitation of other, unrelated proteins. If this would be the case, refolding with RNA would mainly affect these RNA-binding proteins while the other, co-sequestered proteins (“various proteins that engage RNA-binding proteins as well as molecular chaperones”) would be expected to remain insoluble. This is simply not the case as most, if not all, of the proteins refolded in the presence of RNA are soluble – but aggregate again if RNA is removed, see for example Figs 3, 5 and 6.

The Maharana paper from the Hyman/Alberti labs is included and discussed in several places in our manuscript.

We reiterate our argument that while previous studies have described the effect of RNA on single or a few RNA-binding proteins, we suggest in this paper that RNA actually has a much wider activity in modulating protein solubility. These findings are important as they have conceptual and practical implications for several biological fields, as stated before.

2. In the experiment depicted in Fig. 3a, it remains uncertain what types of RNA are contributing to the activity. It remains unclear whether this activity be supported by microRNAs, rRNA, mRNA, long

noncoding RNA etc. Since DNA can have the same effect, this would appear to be a nonspecific chemical chaperone effect of nucleic acids above a certain size (as tRNA is ineffective). It remains unclear whether other nucleic acids above a certain size, such as poly(ADP-ribose) or even other long anionic polymers like polyphosphate or heparin would have the same effect. It thus remains completely uncertain whether there is anything special about RNA in this context.

Fig. 3a outlines the principle of the experiments used in Fig 3 and other experiments which follow. Addition of DNase has no effect on protein aggregation in cell lysate, but RNase does, strongly arguing that RNA is important for maintaining protein solubility in the lysate. Regarding other polyanions, we have performed the same experiment as with RNA and DNA using heparin (page 8, second paragraph and Expanded View Fig. 4a; Discussion page 15, second paragraph). In this experiment, heparin has no effect.

We then use the same assay with short defined DNA oligonucleotides (60-90 nts, similar in size to tRNA) that differ in their sequence and theoretical structure. Here it is clear that oligonucleotides of similar composition (same proportion of purines and pyrimidines) but with different structural configurations have very different capacities to promote the refolding of aggregated proteins (Figures 5 and 6). If this was a nonspecific chemical chaperone effect, one would expect no difference between these oligonucleotides but a difference is obvious.

Based on our RNA sequencing data several types of RNA can maintain protein solubility after denaturation and refolding, including ncRNA and mRNA.

3. In Fig. 3e, the authors attribute stimulated ATPase activity to RNA-mediated refolding. However, in such a crude assay it is not possible to make this conclusion very strongly. It could simply be that RNA stimulates the ATPase activity of a variety of proteins in these crude extracts, which is independent of any effect of refolding, i.e. addition of RNA would stimulate ATPase activity in any crude extract. This possibility is not controlled for and this remains the case despite the rebuttal of the authors. Likewise, it is not clear whether the RNA preparations used here contain any contaminating ATPases. To make this case more strongly, this protocol (i.e. renaturation in the presence of RNA) should be employed on single, purified ATPases (such as any of the ATPases that precipitate upon RNA depletion, e.g. VCP) where the appropriate controls can be readily performed. These issues remain unaddressed and remain unanswered in the revised manuscript.

We do not agree with the referee on this point. Firstly, if the RNA had no effect on the solubility of the proteins, then there should be ATPase activity in the samples without RNA – but this is not the case (Fig 3e and Expanded view Fig 4d). Secondly, our statement that RNA helps to refold ATPases to a functional state should also be viewed in light of the binding of ATPases to ATP-agarose (Fig 4a and Expanded View Spreadsheet 2) which only occurs if the proteins are refolded in the presence of RNA.

4. The authors focus on the properties of nucleic acids that help maintain the solubility of previously aggregated protein after they have been dissolved with 6M GuHCl. However, it is unclear whether there is anything special about RNA (see point 2 above). This activity is very different from nucleic acids directly resolubilizing protein aggregates as the authors appear to suggest throughout the paper (despite changes to the abstract). The nucleic acids depend on 6M GuHCl to first actually solubilize the aggregates, and then merely chaperone (as opposed to disaggregate) the proteins as the GuHCl is removed. This important point is still not evident throughout the text where the authors overstate their findings.

We state on page 7, last paragraph and in Figure 3a that the proteins are first denatured in Guanidine hydrochloride (GuHCl) followed by renaturing with different additives, e.g. RNA, oligonucleotides etc. To make this clearer, we have now altered the heading on page 7, last paragraph to: "RNA is required for maintaining the non-aggregated state of renatured proteins" and added a sentence reading: "We hereafter refer to this process, i.e. GuHCl denaturation of RNase-aggregated proteins followed by refolding in the presence of various additives, as renaturation." We have also modified the first sentence of the second paragraph on page 13 to read: "Finally, we isolated aggregated proteins from brain tissue samples of two patients with ALS and investigated whether NF-H within these could be renatured (after denaturation in GuHCl) with nucleic acids."

5. The application of their protocol to ALS patient samples is not informative as they focus on NF-H, which is of no relevance to the disease. They could have taken this approach with any brain lysate and so it is misleading to suggest that their approach has any relevance to ALS. It continues to remain unclear why they did not assess TDP-43 or other ALS-relevant proteins (e.g. c9-DPRs, FUS, SOD1), which in some cases are extremely abundant proteins.

Our reference to and work on Nf-H and on ALS serve completely different purposes, and we have not suggested that Nf-H is a key pathogenic protein in ALS as are proteins encoded by genes involved in familial ALS (TDP-43; C9orf72 etc). The choice of Nf-H is dictated by the simple fact that it is a protein which is frequently encountered in brain aggregates, particularly in the neurodegenerative process seen in ALS and also in circulation. The choice of ALS as a paradigm of neurodegeneration is because it is a condition characterised by the formation of protein aggregates.

Furthermore, we disagree with the statement " NfH has no relevance to the disease". While, as discussed above, it is clear that NfH is not the product of a causative genetic mutation in ALS, it has been heavily implicated in pathogenesis of the disease. Evidence includes the observation of abnormal NfH subunit accumulation in neuronal perikarya and spheroids in affected spinal cord areas, the risk of developing ALS associated with polymorphisms of the NfH gene and, more importantly, NfH being one the most informative neurochemical markers of the disease whereby its fluid expression has strong specificity and sensitivity in the diagnostic definition of ALS and is linked to rate of progression.

Two references are given below as examples, but there are many others in the literature:

*Lu CH, Petzold A, Topping J, Allen K, Macdonald-Wallis C, Clarke J, Pearce N, Kuhle J, Giovannoni G, Fratta P, Sidle K, Fish M, Orrell R, Howard R, Greensmith L, Malaspina A. Plasma neurofilament heavy chain levels and disease progression in amyotrophic lateral sclerosis: insights from a longitudinal study. *J Neurol Neurosurg Psychiatry*. 2015 May;86(5):565-73. doi: 10.1136/jnnp-2014-307672.*

*Xu Z, Henderson RD, David M, McCombe PA. Neurofilaments as Biomarkers for Amyotrophic Lateral Sclerosis: A Systematic Review and Meta-Analysis. *PLoS One*. 2016 Oct 12;11(10):e0164625. doi: 10.1371/journal.pone.0164625.*

6. They continue to ignore several key papers from the literature perhaps to make their work seem more novel. For example, several excellent papers are not cited including: Sulijoadikusumo et al., 2001 and Sun et al., 2014.

We are surprised by the extraordinary allegation that we are deliberately ignoring key papers so that our work seems more novel.

However, we have now included all nine papers suggested by the Referee except for Elden et al, Nature (2010) as it is not relevant. The Elden paper shows that RNase treatment abolishes interaction between Ataxin-2 and TDP-43, but does not consider protein aggregation. In addition, we have included Swanson and Dreyfuss, EMBO J (1988) which describes RNA binding of hnRNP proteins – see Discussion, page 13, first paragraph.

7. The authors still do not seem to understand the difference between protein aggregation and liquid-liquid phase separation.

We believe that this refers to the Introduction where we touch upon how phase-separation and protein aggregation could be linked. There are several excellent reports and reviews substantiating such a connection. These include Lin et al., 2015 and Elbaum-Garfinkle 2019, and others which we have now described in the revised Introduction, page 3. We would therefore argue that this information is relevant in its current form.

References:

Lin et al., Formation and Maturation of Phase-Separated Liquid Droplets by RNA-Binding Proteins. *Molecular Cell*, 2015, 60: 189-192

Elbaum-Garfinkle, Matter over mind: Liquid phase separation and neurodegeneration. *J Biol Chem*. 2019 3;294(18):7160-7168

Referee #2:

The authors have addressed all the concerns and the paper should be published.

Referee #3:

Aarum et al reports here an interesting studies showing that RNA degradation causes precipitation of proteins. I've read the manuscript and the provided reviews from referee 1 and 2. See my points below:

1. Which proportion of the protein precipitates. I searched for information across the text that could provide any clue about the penetrance of the effect but this info is not available (or I couldn't find it). Which % of the total protein precipitates? This should be disclosed in all figures in the text to differentiate between a real major effect and a marginal consequence of loss of RNA.

The amount of aggregated proteins is 10% ± 1 (n=4) of the total amount of soluble proteins in the starting material. This information has been added to the Results, page 5, end of first paragraph. We found this proportion to be highly consistent, even after prolonged (several hours) of RNase treatment and irrespective of the amount of starting material.

2. Looking at the silver staining in figure 1, I don't see major differences in profile from the input and the pellet. This would be expected if the precipitation would be biased toward a given pool of proteins, e.g. RBPs (see RBP silver staining pattern in Baltz et al. Mol Cell 2012 and Castello et al Cell 2012). This is supported by the precipitation of ACTB, which is an abundant protein that, to my knowledge, does not interact with RNA.

We appreciate that this is not very clear and believe this is due to the relatively large amount, both in terms of mass and numbers, of proteins present in the pellet and supernatant together with the

resolution of the Coomassie gel where proteins of similar size and amount will be difficult to distinguish between lanes. Similar results have been obtained by others using a comparable approach (b-isox which precipitates proteins with low complexity regions, Kato et al., Cell. 2012, 149(4): 753). This has now been considered in the Discussion, bottom of page 18 and top of page 19.

3. The proteomic analysis is biased. Authors only analysed (to my knowledge) the pellet. If you mass spec a whole cell lysate in a shotgun approach, near 30% of the identified proteins (normally from 3000 to 5000) will be RBPs just because many of the housekeeping RBPs (ribosomal protein, hnRNPs, etc) are very abundant. For this experiment to be informative it is necessary to analyse the input and the supernatant and perform the enrichment analyses against these datasets. By doing so, it would be possible to correct these biases and provide new cues about the proteins within the whole cell proteome that exhibit a differential behaviour in absence of RNA. If authors are not able to perform such experiments, at least, should use a whole proteome similar in size and generation conditions in a suitable line. There are a few made in HEK293 and HeLa.

We thank the referee for raising this point. We unfortunately did not analyse the input nor the supernatant. The initial enrichment-analysis was performed against the whole theoretical human proteome which we agree is biased. However, we have now re-analysed the data against the whole proteome from a similar cell type to what we used, day 15 differentiated human neurons derived from neural stem cells, as suggested by the referee. We have also analysed any enrichment of RNase-aggregated proteins against the most abundant proteins in this reference set. The findings from the new analyses are described in Results, page 6, first paragraph, and in Expanded view figure 2 and in Expanded view spreadsheet 1.

4. I agree with referee 2 that authors should highlight better in what points this work differ from previous related works. I see a strong point is the proteomics, however, see my points in comment 3.

We have addressed this point in the Introduction, end of page 4, and in the Discussion, end of page 13 and elsewhere.

Dear Denise,

Thank you for your patience while your revised manuscript was peer-reviewed. We have now received comments from referees 1 and 3 as well as cross-comments, and all are pasted below.

As you will see, while referee 1 remains unconvinced, referee 3 thinks that the revised manuscript can be published. This is a borderline case. Given the cross-comments by referee 3, we have decided that we can offer to publish your study, if all remaining concerns can be satisfactorily addressed, at least in the manuscript text. Please also re-submit a detailed point-by-point response to all remaining comments. Please do let me know whether a reconstitution assay with the VCP ATPase could be performed, and please do include and discuss TDP-43 data you (might) have. Please also state more explicitly what exactly the novelty of your study is. I would also like to suggest to add more specific information to the abstract; what/how much is widespread, global and many? How many proteins/what percentage are affected? What cells or tissues were used?

A few more minor changes will also be required:

Some figures are in landscape format, please change to portrait.

The figure resolution needs to be increased throughout to meet production quality.

All figures (main and EV) need to be uploaded as separate files. Table EV1 needs to be uploaded as individual file as well.

The Expanded View figures are incorrectly called out. Please use 'Figure EV#' or 'Table EV#'.

Please add file names to the DATASETS. (The legends are in the files, but the files names are not). They should contain a header 'Dataset EV#'.

Table EV1 is incorrectly called out, and needs to be removed from the Article file.

Please upload the movie as ZIPed file including the legend, which needs to be removed from the Article file.

We can only offer a maximum of 5 EV figures, and the manuscript currently has 8. May be these could be combined to 5. Otherwise all extra figures could be moved to an Appendix file. Please see our guide to authors for more information.

Please add a Data Availability Section at the end of your materials and methods that lists the GEO accession number and a direct link to the website with the deposited data.

I would like to suggest a minor change to the abstract (and please be more specific on how many proteins are affected, and which cells or tissues were used, as noted above). Please let me know whether you agree with the following:

Most proteins in cell and tissue lysates are soluble. Here, we show that many of these proteins, including several that are implicated in neurodegenerative diseases, are maintained in a soluble and functional state by association with endogenous RNA, as degradation of RNA invariably leads to protein aggregation. We identify the importance of nucleic acid structure, with single-stranded

pyrimidine-rich bulges or loops surrounded by double-stranded regions being particularly efficient in this role, revealing an apparent one-to-one protein-nucleic acid stoichiometry. We also show that protein aggregates isolated from brain tissue from Amyotrophic Lateral Sclerosis patients can be rendered soluble after refolding by both RNA and synthetic oligonucleotides. Together, these findings open new avenues for understanding the mechanism behind protein aggregation and shed light on how certain proteins remain soluble.

EMBO press papers are accompanied online by A) a short (1-2 sentences) summary of the findings and their significance, B) 2-3 bullet points highlighting key results and C) a synopsis image that is 550x200-400 pixels large (the height is variable). You can either show a model or key data in the synopsis image. Please note that text needs to be readable at the final size. Please send us this information along with the revised manuscript.

I attach to this email a related manuscript file with comments by our data editors. Please address all comments in the final manuscript file.

Kind regards,
Esther

Referee #1:

In this second revision, Sheer and colleagues have continued to not address my prior concerns. The major issue with this work is novelty. It is simply not surprising that RNA can chaperone proteins, the precedent has long been set with a large number of individual proteins, including model substrates that do not bind RNA. That this concept applies to many other proteins is simply not surprising. This issue alone makes the work unsuitable for EMBO Reports where novelty is important.

Several assays remain very crude (e.g. attribute stimulated ATPase activity to RNA-mediated refolding), which could be explained in many other ways. Some reconstitution with a specific ATPase such as VCP would be needed to provide definitive evidence but this is still missing.

It remains unclear why the authors have not studied TDP-43 in their ALS patient samples. Nf-H is not very relevant to ALS despite claims that it may be a biomarker. One suspects that TDP-43 is not solubilized by RNA in contrast to the broad claims made by the authors and so they are deliberately omitting these important data.

Bafflingly, the authors think that the study by Elden et al. is relevant. In that paper, it is shown that mutating TDP-43 RRM1 to forms that cannot engage RNA makes TDP-43 more aggregation-prone in cells. This finding is simply another result that reduces the novelty of the present work.

Overall, in my view, this work is not suitable for EMBO Reports and should be submitted to a more

specialized journal.

Referee #3:

Authors have done a reasonable work answering my comments questions. They employ an already available proteome to perform enrichment analysis against the right background. Obviously, the optimal scenario would be to have analysed the input of the experiment. However, given the inability to perform experiments due to current pandemic, it is a good compromise and the data is now cleaner. Hence, I am satisfied with the additions and I think that the work have overall improved substantially.

Cross-comments from referee 3:

Here my comments:

It is simply not surprising that RNA can chaperone proteins, the precedent has long been set with a large number of individual proteins, including model substrates that do not bind RNA. That this concept applies to many other proteins is simply not surprising. This issue alone makes the work unsuitable for EMBO Reports where novelty is important.

- Well, I am unsure if 'unexpected' or 'expected' are reasons to reject a paper. In this way, researchers will just not follow any path that may lead to an 'expected' outcome. While I agree that the concept of 'aggregation in absence of RNA' per se is not new, authors extend an idea that has been limited to a very few example to a large population of proteins within the cell. I personally think that an exciting result should be 'novel' but not necessarily 'surprising' to make it exciting. I think anyway this belongs to a subjective area that has very little of scientific. It is a matter of whether the authors have highlighted the novelty of their discoveries in the present manuscript.

Several assays remain very crude (e.g. attribute stimulated ATPase activity to RNA-mediated refolding), which could be explained in many other ways. Some reconstitution with a specific ATPase such as VCP would be needed to provide definitive evidence but this is still missing.

- I agree with this point. However, EMBO reports are short manuscripts that, to my knowledge, rely more on the novelty of the discovery than in the full mechanistic characterisation (which is more typical in EMBO J). The experiment that the referee request is a full project by itself and I think is beyond the scope of this paper. The concept that authors try to convey is that RNA is an important molecule at solubilising a substantial part of the proteome.

It remains unclear why the authors have not studied TDP-43 in their ALS patient samples. Nf-H is not very relevant to ALS despite claims that it may be a biomarker. One suspects that TDP-43 is not solubilized by RNA in contrast to the broad claims made by the authors and so they are deliberately omitting these important data.

I am not aware of the details of these data as it is not the focus of my research. However, I have the impression that it is not central to the paper. Authors could refer to this and contextualise their results if the referee clarifies to which paper does he/she refers to.

Bafflingly, the authors think that the study by Elden et al. is relevant. In that paper, it is shown that mutating TDP-43 RRM2 to forms that cannot engage RNA makes TDP-43 more aggregation-prone in cells. This finding is simply another result that reduces the novelty of the present work.

As mentioned above, this paper is not about individual examples. It is about expanding a concept to a large subpopulation of the cellular proteome. In this context TDP-43 is one example.

I agree this is a complicated borderline case. I think the concept is interesting because again expands a concept that was limited to few proteins to a substantial proteome population. It is clear that the field is very interested on the mechanisms behind phase transitions and the formation of membrane bound organelles, however, the importance of RNA in the process is normally omitted. This paper provide evidence that RNA may be an important component in the equation as in its absence, RBPs may precipitate and form aggregates in the cell. So the final point is whether expanding a concept is important enough and whether authors have highlighted the novelty of their discoveries effectively.

EMBOR-2019-49585V3_Response to Referees

We thank the referees for their reviews of our manuscript.

Referee #1:

In this second revision, Sheer and colleagues have continued to not address my prior concerns. The major issue with this work is novelty. It is simply not surprising that RNA can chaperone proteins, the precedent has long been set with a large number of individual proteins, including model substrates that do not bind RNA. That this concept applies to many other proteins is simply not surprising. This issue alone makes the work unsuitable for EMBO Reports where novelty is important.

Several assays remain very crude (e.g. attribute stimulated ATPase activity to RNA-mediated refolding), which could be explained in many other ways. Some reconstitution with a specific ATPase such as VCP would be needed to provide definitive evidence but this is still missing.

- *As stated previously, the purpose of the study was to investigate how widespread the solubilizing effect of RNA is on a large number of proteins, not to investigate single mechanisms of individual proteins in detail. We believe that it is the principle that is important here, i.e. does degradation of RNA in soluble cell extracts result in protein aggregation and can RNA help to maintain protein solubility after re-folding, and are the proteins functional afterwards? Assaying individual proteins such as VCP is thus beyond the scope of our manuscript. We also reiterate that the ATPase experiment must be viewed in context of the other findings, e.g. the binding of re-natured proteins to immobilized ATP.*

It remains unclear why the authors have not studied TDP-43 in their ALS patient samples. Nf-H is not very relevant to ALS despite claims that it may be a biomarker. One suspects that TDP-43 is not solubilized by RNA in contrast to the broad claims made by the authors and so they are deliberately omitting these important data.

- *The whole purpose of the experiment on ALS brain samples was to highlight the principle of our findings on appropriate human tissue. No relevant data has been purposely omitted from the manuscript. The reason we do not use TDP-43 is that we could not detect any in the starting material (brain lysate), despite using several antibodies. Since the material was limited, we therefore decided to use Nf-H which was readily detected, and despite what the referee claims, is present in ALS protein aggregates, see references below. A paragraph describing this, including references, is now included in the manuscript, page 13, second section.*

References:

- *Didonna A, Opal P (2019) The role of neurofilament aggregation in neurodegeneration: lessons from rare inherited neurological disorders. Molecular Neurodegeneration 14: 19*
- *Mendonça DMF, Chimelli L, Martinez AMB (2005) Quantitative evidence for neurofilament heavy subunit aggregation in motor neurons of spinal cords of patients with amyotrophic lateral sclerosis. Brazilian Journal of Medical and Biological Research 38: 925-933*

Bafflingly, the authors think that the study by Elden et al. is relevant. In that paper, it is shown that mutating TDP-43 RRMs to forms that cannot engage RNA makes TDP-43 more aggregation-prone in cells. This finding is simply another result that reduces the novelty of the present work.

The Elden paper shows that interaction between Ataxin-2 and TDP-43 is dependent on RNA. It does, however, also show that RRM-mutated TDP-43, which cannot bind RNA, forms cytoplasmic aggregates when excluded from the nucleus. We have therefore cited the Elden study in our paper on page 8, end of third paragraph, together with Pesiridis et al., which similarly shows that TDP-43 aggregation is enhanced by lack of RNA interaction.

Overall, in my view, this work is not suitable for EMBO Reports and should be submitted to a more specialized journal.

Referee #3:

Authors have done a reasonable work answering my comments questions. They employ an already available proteome to perform enrichment analysis against the right background. Obviously, the optimal scenario would be to have analysed the input of the experiment. However, given the inability to perform experiments due to current pandemic, it is a good compromise and the data is now cleaner. Hence, I am satisfied with the additions and I think that the work have overall improved substantially.

- *We thank the referee for his/her comments and suggestions.*

Cross-comments from referee 3:

[These are comments on the review above from referee 1]

Here my comments:

It is simply not surprising that RNA can chaperone proteins, the precedent has long been set with a large number of individual proteins, including model substrates that do not bind RNA. That this concept applies to many other proteins is simply not surprising. This issue alone makes the work unsuitable for EMBO Reports where novelty is important.

- Well, I am unsure if 'unexpected' or 'expected' are reasons to reject a paper. In this way, researchers will just not follow any path that may lead to an 'expected' outcome. While I agree that the concept of 'aggregation in absence of RNA' per se is not new, authors extend an idea that has been limited to a very few example to a large population of proteins within the cell. I personally think that an exciting result should be 'novel' but not necessarily 'surprising' to make it exciting. I think anyway this belongs to a subjective area that has very little of scientific. It is a matter of whether the authors have highlighted the novelty of their discoveries in the present manuscript.

Several assays remain very crude (e.g. attribute stimulated ATPase activity to RNA-mediated refolding), which could be explained in many other ways. Some reconstitution with a specific ATPase

such as VCP would be needed to provide definitive evidence but this is still missing.

- I agree with this point. However, EMBO reports are short manuscripts that, to my knowledge, rely more on the novelty of the discovery than in the full mechanistic characterisation (which is more typical in EMBO J). The experiment that the referee request is a full project by itself and I think is beyond the scope of this paper. The concept that authors try to convey is that RNA is an important molecule at solubilising a substantial part of the proteome.

It remains unclear why the authors have not studied TDP-43 in their ALS patient samples. Nf-H is not very relevant to ALS despite claims that it may be a biomarker. One suspects that TDP-43 is not solubilized by RNA in contrast to the broad claims made by the authors and so they are deliberately omitting these important data.

- I am not aware of the details of these data as it is not the focus of my research. However, I have the impression that it is not central to the paper. Authors could refer to this and contextualise their results if the referee clarifies to which paper does he/she refers to.

- *As mentioned above, we have added a paragraph describing our choice of NF-H, page 13, second section, together with the two references shown above:*

Didonna A, Opal P (2019)

Mendonça DMF, Chimelli L, Martinez AMB (2005)

Bafflingly, the authors think that the study by Elden et al. is relevant. In that paper, it is shown that mutating TDP-43 RRMs to forms that cannot engage RNA makes TDP-43 more aggregation-prone in cells. This finding is simply another result that reduces the novelty of the present work.

- As mentioned above, this paper is not about individual examples. It is about expanding a concept to a large subpopulation of the cellular proteome. In this context TDP-43 is one example.

I agree this is a complicated borderline case. I think the concept is interesting because again expands a concept that was limited to few proteins to a substantial proteome population. It is clear that the field is very interested on the mechanisms behind phase transitions and the formation of membrane bound organelles, however, the importance of RNA in the process is normally omitted. This paper provide evidence that RNA may be an important component in the equation as in its absence, RBPs may precipitate and form aggregates in the cell. So the final point is whether expanding a concept is important enough and whether authors have highlighted the novelty of their discoveries effectively.

Prof. Denise Sheer
Queen Mary University of London
Blizard Institute
Barts and The London School of Medicine and Dentistry
4 Newark Street
London, London E1 2AT
United Kingdom

Dear Denise,

I am very pleased to accept your manuscript for publication in the next available issue of EMBO reports. Thank you for your contribution to our journal.

At the end of this email I include important information about how to proceed. Please ensure that you take the time to read the information and complete and return the necessary forms to allow us to publish your manuscript as quickly as possible.

As part of the EMBO publication's Transparent Editorial Process, EMBO reports publishes online a Review Process File to accompany accepted manuscripts. As you are aware, this File will be published in conjunction with your paper and will include the referee reports, your point-by-point response and all pertinent correspondence relating to the manuscript.

If you do NOT want this File to be published, please inform the editorial office within 2 days, if you have not done so already, otherwise the File will be published by default [contact: emboreports@embo.org]. If you do opt out, the Review Process File link will point to the following statement: "No Review Process File is available with this article, as the authors have chosen not to make the review process public in this case."

Should you be planning a Press Release on your article, please get in contact with emboreports@wiley.com as early as possible, in order to coordinate publication and release dates.

Thank you again for your contribution to EMBO reports and congratulations on a successful publication. Please consider us again in the future for your most exciting work.

THINGS TO DO NOW:

You will receive proofs by e-mail approximately 2-3 weeks after all relevant files have been sent to our Production Office; you should return your corrections within 2 days of receiving the proofs.

Please inform us if there is likely to be any difficulty in reaching you at the above address at that time. Failure to meet our deadlines may result in a delay of publication, or publication without your corrections.

All further communications concerning your paper should quote reference number EMBOR-2019-49585V4 and be addressed to emboreports@wiley.com.

Should you be planning a Press Release on your article, please get in contact with emboreports@wiley.com as early as possible, in order to coordinate publication and release dates.

Corresponding Author Name: Johan Aarum and Denise Sheer

Manuscript Number: EMBOR-2019-49585V3